# Atomic Layer Deposition of Antibacterial Nanocoatings: A Review

**DOI:** 10.3390/antibiotics12121656

**Published:** 2023-11-24

**Authors:** Denis Nazarov, Lada Kozlova, Elizaveta Rogacheva, Ludmila Kraeva, Maxim Maximov

**Affiliations:** 1Peter the Great Saint Petersburg Polytechnic University, Polytechnicheskaya, 29, 195221 Saint Petersburg, Russia; maximspbstu@mail.ru; 2Saint Petersburg State University, Universitetskaya Nab, 7/9, 199034 Saint Petersburg, Russia; lada.kozlova.20@mail.ru; 3Saint-Petersburg Pasteur Institute of Epidemiology and Microbiology, 14 Mira Street, 197101 Saint Petersburg, Russia; elizvla@yandex.ru (E.R.); lykraeva@yandex.ru (L.K.)

**Keywords:** atomic layer deposition, antibacterial coatings, zinc oxide, titanium oxide, silver coatings, medical implants

## Abstract

In recent years, antibacterial coatings have become an important approach in the global fight against bacterial pathogens. Developments in materials science, chemistry, and biochemistry have led to a plethora of materials and chemical compounds that have the potential to create antibacterial coatings. However, insufficient attention has been paid to the analysis of the techniques and technologies used to apply these coatings. Among the various inorganic coating techniques, atomic layer deposition (ALD) is worthy of note. It enables the successful synthesis of high-purity inorganic nanocoatings on surfaces of complex shape and topography, while also providing precise control over their thickness and composition. ALD has various industrial applications, but its practical application in medicine is still limited. In recent years, a considerable number of papers have been published on the proposed use of thin films and coatings produced via ALD in medicine, notably those with antibacterial properties. The aim of this paper is to carefully evaluate and analyze the relevant literature on this topic. Simple oxide coatings, including TiO_2_, ZnO, Fe_2_O_3_, MgO, and ZrO_2_, were examined, as well as coatings containing metal nanoparticles such as Ag, Cu, Pt, and Au, and mixed systems such as TiO_2_-ZnO, TiO_2_-ZrO_2_, ZnO-Al_2_O_3_, TiO_2_-Ag, and ZnO-Ag. Through comparative analysis, we have been able to draw conclusions on the effectiveness of various antibacterial coatings of different compositions, including key characteristics such as thickness, morphology, and crystal structure. The use of ALD in the development of antibacterial coatings for various applications was analyzed. Furthermore, assumptions were made about the most promising areas of development. The final section provides a comparison of different coatings, as well as the advantages, disadvantages, and prospects of using ALD for the industrial production of antibacterial coatings.

## 1. Introduction

Different nanomaterials can provide strong bactericidal and bacteriostatic effect, i.e., reduce the adhesion of individual bacteria and the rate of colony formation, as well as inhibit their proliferation [1,2,3]. The global COVID-19 pandemic has dramatically increased interest in and demand for more effective antimicrobial and antiviral nanocoatings [4,5]. It has also sparked a wave of innovation. Innovators are using a wide range of approaches, including traditional biocides, photocatalytic coatings, and new nanocoatings. In addition, research is actively underway to improve processes for obtaining known antibacterial nanocoatings.

To date, a wide variety of methods have been used to produce antibacterial nanocoatings. These deposition techniques use different physical and chemical processes and have been reviewed in a number of publications [5,6,7]. Among these technologies, atomic layer deposition (ALD) is particularly noteworthy. ALD is a successfully developing industrial technology, but it is mainly used in microelectronics [8,9], Li-ion battery modification [10,11], sensors [12,13], and catalysis [14,15], and it has not yet been practically applied in the medical industry. At the same time, a significant number of papers have been published to date in which ALD has been successfully applied to antibacterial coatings. In recent years, there has been a crisis in the reproducibility of scientific publications [16,17], whereas ALD is a standardized and automated method and its results are usually well reproducible in most commercial and handmade equipment [18,19]. Therefore, a literature review on the synthesis of antibacterial coatings via ALD can provide a fairly reliable comparative analysis of the results obtained by different research groups and thus provide new reliable conclusions.

ALD is a method of synthesizing inorganic thin films and coatings based on the reaction of chemical groups on the substrate surface with reagents in the gas phase [20,21]. A special feature of the method is the self-saturating nature of such reactions, as the reactive groups on the substrate surface are depleted. Once saturation is reached, these groups can be recovered by the next chemical reaction. The self-saturation and recovery processes can be carried out cyclically, allowing a controlled build-up of the coating with high thickness accuracy (Figure 1). Since chemical reactions and film growth occur only at the substrate–gas phase interface, the resulting coatings are characterized by high thickness uniformity, even on substrates with complex topography and porous substrates (conformality) (Figure 1) [22]. Because the process is based on chemical reactions with the substrate surface, the resulting coatings are chemically bonded to the substrate and therefore have high adhesion. Another important feature of ALD is the ability to run the process at relatively low temperatures, in some cases close to room temperature, which allows the coating of temperature sensitive substrates such as polymers, fabrics, and various biological substrates.

Due to the nature of the chemical processes, ALD is best suited for the synthesis of oxides and their mixtures or nanolaminates. Metals such as Ag, Cu, Au, Ni, and Pt can also be produced via ALD, but due to the peculiarities of nucleation, individual nanoparticles usually grow rather than a continuous film growing [23,24]. One of the disadvantages of this method is the low growth rate. Since only a fraction of a monomolecular layer grows during an ALD cycle of several seconds to several minutes, the normal growth rate for most ALD processes does not exceed tens of nanometers per hour. Therefore, ultrathin films with thicknesses ranging from a few nanometers to tens of nanometers, and in a few cases up to several hundred nanometers, are typically obtained via ALD.

Despite the low growth rate, nanocoatings of inorganic bactericidal materials obtained via ALD can find wide applications in a variety of products such as dental and orthopedic implants [25,26,27,28], contact lenses [29], touch screens [30,31,32], antibacterial textiles [33,34,35,36], and water purification systems [37,38,39].

Microbiocidal and microbiostatic effects are possible for inorganic oxide and metallic nanoscale coatings. The microbicidal effect is achieved by various mechanisms such as the generation of reactive oxygen species (ROS), ion permeation, and the “contact-killing” mechanism. ROS such as hydrogen peroxide (H_2_O_2_), superoxide (O_2_^−^), hydroxyl radical (^•^OH), and singlet oxygen (^1^O_2_) cause oxidative stress by directly damaging the lipids of bacterial cell membranes, resulting in the loss of membrane function [40]. Ions of many inorganic compounds can penetrate bacterial membranes and disrupt metabolic processes and ribonucleic acid (RNA) replication [41]. In addition, some nanocoatings, due to their special morphology and surface topography, can disrupt the integrity of bacterial cell walls and implement the “contact-killing” mechanism.

The influence of surface hydrophilicity should also be considered when analyzing antibacterial properties. Unfortunately, it is not possible to say unequivocally which level of hydrophilicity/hydrophobicity is ideal for the best antibacterial properties. The optimum surface hydrophilicity depends on the application. For example, in the case of medical implant coatings, a more hydrophilic surface favors the sorption of various peptides and biomolecules, host cells, and bone extracellular matrix, and thus microbial pathogens lose the “race for the surface” [42]. On the other hand, hydrogen bonding and weak hydrophobic interactions between bacterial peptidoglycan and the surface can easily facilitate bacterial adhesion on a moderately wettable surface [41]. Surface charge is also important, as bacterial surfaces are usually negatively charged and a positive surface charge of coatings favors bacterial adhesion [41]. In addition, surface morphology and topography can significantly influence bacterial cell adhesion, as a surface with a different morphology provides different conditions in terms of the size and number of contact sites for bacteria.

At very small thicknesses, typical for ALD coatings, the influence of the substrate on wettability, structure, and defects is manifested. In addition, due to the significant share of the defective near-surface layer in the total volume of nanocoatings, their thickness largely determines their functional properties, including their antibacterial activity.

Therefore, in order to analyze the antibacterial effect of different coatings, not only should the chemical composition of the coating be taken into account, but also its thickness [43], topography and morphology, hydrophilicity, and surface charge [44], and, in the case of the photocatalytic mechanism of ROS generation, the crystalline structure of the coatings is fundamental [45].

In this review, we provide a detailed analysis of the use of various inorganic coatings produced via ALD as antibacterial coatings. The results section of the review is structured according to the chemical composition of the coatings, but the following sections also provide a discussion of the results, with particular emphasis on areas of practical application, comparing the results for different compounds and analyzing the influence of the above factors on the antibacterial performance. Finally, in the last section, an attempt is made to evaluate the future prospects of ALD.

## 2. Methods

### Literature Search

The comprehensive electronic search was carried out using the Scopus database. The search and analysis of the data was partly conducted using PRISMA (Preferred Reporting Items for Systematic Review and Meta-Analyses) [46]. The Appendix A has a thorough explanation of how we searched the literature, determined which materials to include or exclude, and analyzed the data.

## 3. Results

### 3.1. Titanium Oxide

#### 3.1.1. Photocatalytic Generation of ROS

Despite the ease of obtaining titanium oxide via ALD, there are not many studies on its antibacterial properties (Table 1). The antibacterial activity of titanium oxide is strongly dependent on the influence of near ultraviolet (UV) light on the material. TiO_2_ is a semiconductor with a band gap of about 3.2 eV and therefore electron–hole pairs are formed under UV irradiation with a wavelength of less than 380 nm. In the presence of water, the conduction band electrons can react with molecular oxygen to form superoxide anions (O_2_^−^) but the valence band holes oxidize water to hydroxyl radicals (^•^OH) [47] (Figure 2a). These ROS can readily break down organic matter, disrupting the cell walls of bacteria and killing them.

The disruption of cell wall integrity due to ROS action can be observed via scanning electron microscopy. For example, Zhang et al. observed a significant bactericidal effect due to cell membrane damage of *E. coli* on the surface of TiO_2_ nanoarrays after sunlight irradiation (Figure 2b–e) [39]. In fact, according to a number of studies [29,30,39,54], significant antibacterial and antifungal activity is only shown when TiO_2_ is irradiated in the ultraviolet range. According to the results of Godlewski et al., who tested various ALD oxides on 16 different strains without additional illumination, TiO_2_ has significantly less bactericidal activity than not only ZnO but also ZrO_2_, HfO_2_, and Al_2_O_3_ [55]. The question of which types of ROS are the most active and play a more important role in the antibacterial effect of remains open, but Zhang et al. showed that hydroxyl radicals are predominantly formed and determine the activity against *E. coli* (Figure 2f–h) for TiO_2_ nanoarrays obtained using combination ALD and solvothermal deposition [39].

However, even when UV radiation is used, the results obtained by different authors vary considerably. The photocatalytic activity of TiO_2_ is strongly dependent on impurities and structural defects, which are particularly abundant in nanocoatings. Thermally generated defects of TiO_2_ could reduce the lifetime of the photogenerated electron–hole pairs, so the use of low temperature processes can help to reduce these defects [56]. In addition, the thickness of the coatings and the specific surface area can be important factors in increasing the number of ROS formed. Absorption of UV radiation extends to micron thicknesses but diffusion of ROS to the surface is only limited by the surface layer of a few nanometers (Figure 3) [47,50]. For ultrathin nanofilms, the substrate material also plays an important role, as the efficiency of photocatalytic antibacterial surfaces is limited by the absorption of light in it [47,50].

However, there is evidence that nanocoatings can generate ROS even without UV irradiation because they have many defects in their structure that add new levels to the forbidden band of the material. The antibacterial and antifungal activity of TiO_2_ under dark measurements has indeed been recorded in many studies [29,39,49,52,53,57,58]. However, the most likely cause of this phenomenon is a change in wettability or other surface properties after coating. In most of these studies, the authors observed an increase in the hydrophilicity of the surface, which leads to the realization of the microbiostatic effect, i.e., a decrease in the adhesion of bacteria and their ability to form a biofilm. This effect is particularly evident for coatings applied to polymeric substrates such as polytetrafluoroethylene (PTFE) [39], polyamide 66 [58], and PDMS [52]; the initial surface of which is hydrophobic and with the application of coatings is significantly hydrophilized.

#### 3.1.2. Morphology and Topography of Surface

Surface morphology and topography can also be a critical factor in determining antibacterial properties. Surfaces with a certain morphology can cause significant stretching of the bacterial cell membrane, resulting in piercing (Figure 4a). Such a disruption of cell membranes was observed by Singh et al. on samples of black silicon nanopillars coated with TiO_2_ using ALD, spin coating, and spray coating (Figure 4e–p). In addition to the contact-killing effect, the surface morphology can significantly influence or even determine the type of bacterial adhesion by changing the area of the contact zones or the charge and wetting of the surface (Figure 4b–d). For example, the TiO_2_-coated nanoporous alumina membrane with a pore size of 20 nm inhibited the adhesion of *S. aureus* and *E. coli*, whereas the TiO_2_-coated membrane with a pore size of 100 nm did not [51]. A significant influence of surface morphology has also been reported in [37] where smooth ALD coatings, nanotubes, and TiO_2_ nanoparticles were compared. Nanomaterials with more developed surface morphology showed better antibacterial activity against *E. coli*.

#### 3.1.3. Thickness of the Coatings

As discussed in Section 3.1.1, film thickness plays a significant role in ROS generation. At present, the literature presents results for ALD of TiO_2_ films with thicknesses ranging from 4 to 200 nm. It is difficult to compare results from different papers, but a number of researchers have considered the influence of thickness experimentally. It has been shown in [50] that the maximum degradation of methylene blue, which is an effective indicator of ROS generation, is achieved for titanium oxide on flat silicon at 40 nm and further increases in thickness have no effect (Figure 5a), as ROS are quite active and their diffusion is only possible from a thin near-surface layer (Figure 3a).

In the study Pessoa et al., TiO_2_ coatings of 50 to 200 nm thickness (500–2000 ALD cycles) on PU and PDMS surfaces were investigated against *C. albicans* yeasts and no noticeable difference was observed after UV exposure (Figure 5b) [29]. In the absence of light, the effect of thickness was recorded, but it was due to changes in surface free energy (SFE) and surface wettability. Comparative studies of lower thickness films were only carried out in one study, but it did not reveal any useful information on the thickness effect: 10 nm, 20 nm and 50 nm ALD TiO_2_-coated polyamide-66 (PA66) fabrics were investigated by Akyildiz et al. using the agar diffusion plate test with *E. coli* and *S. aureus* in both dark and sunlight exposure. All TiO_2_-coated fabrics showed no inhibition zone for both *S. aureus* and *E. coli*. However, the TiO_2_-coated samples were less covered with bacteria compared to the uncoated samples [58].

#### 3.1.4. Crystalline Structure of the Coatings

Titanium oxide has three crystalline modifications (anatase, rutile, brukite), but rutile and brukite are only formed at relatively high temperatures. In ALD temperature regimes, only anatase is usually obtained, which is considered to be much more antibacterial than rutile [45]. However, crystallite formation is only possible from a thickness of about 10 to 12 nm [59], below which the film becomes amorphous regardless of the synthesis temperature. Lee et al. showed that increasing the synthesis temperature (from 70 to 300 °C), and consequently the crystallinity, improves the antibacterial properties against *E. coli* for TiO_2_ under UV exposure [54]. Molina-Reyes et al. also showed that a 10 nm TiO_2_ film with anatase structure deposited on amorphous TiO_2_ nanotubes significantly reduced the survival rate of *E. coli* [45]. However, measurements without additional illumination using *S. aureus* showed that crystalline coatings obtained at 190 °C were even less antibacterial than amorphous coatings obtained at 160 and 120 °C, and when tested on *E. Coli* there was no difference between the samples [49].

However, one of the most interesting effects of crystallinity on antibacterial properties is the synergistic effect of anatase and rutile, which significantly increases bactericidal activity due to the intrinsic alignment of the valence and conduction bands between the anatase and rutile phases. This alignment promotes the separation of photogenerated electrons and holes and increases their activity to generate ROS [45].

#### 3.1.5. Precursors and ALD Temperature

Three main titanium precursors, titanium (IV) isopropoxide (TTIP) [54,60], titanium chloride—TiCl_4_ [29,48,51], and tetrakis(dimethylamido)titanium, TDMAT [34,45,49,52,53] are used for the synthesis of TiO_2_ via ALD. The second reagent used is mainly water, but Darwish et al. used ozone [53] and Goldfinger et al. used oxygen plasma coupled with TDMAT to modify PMMA and PDMS, respectively [52]. At the same time, coating with such a strong oxidant resulted in a significant decrease in the wetting angle (from 110 to <20), whereas modifying the PDMS surface with TiCl_4_/H_2_O, even with rather thick films of 100–200 nm thickness, practically does not change the wettability [29]. At the same time, the use of TiCl_4_/H_2_O has a significant disadvantage—chlorine contamination during synthesis at low temperatures (around 80 °C) [29]. In the synthesis of complex oxide systems, there is also the possibility of chlorine contamination [61]. Titanium oxide synthesis temperatures can vary considerably. When using temperature sensitive substrates, temperatures from 65 °C [53] can be used, and, if required, crystalline coatings can be obtained up to 300 °C [48,51] and higher.

#### 3.1.6. Effect of Coatings on Different Strains

The vast majority of studies on ALD TiO_2_ coatings have been carried out on *E. coli* and *S. aureus*, including methicillin-resistant *S. aureus* (MRSA). Titanium oxide coatings of 4.3 and 8.6 nm thickness deposited on nanoporous alumina membranes are more effective against *S. aureus* than against *E. coli* [51]; however, Liu et al. showed that 100 nm thick films deposited on titanium at different temperatures (120, 160, 190 °C) have higher activity against *E. coli* than against *S. aureus* and MRSA [49]. However, most studies show no significant difference in activity against *S. aureus* and *E. coli* [30,55,58,62].

The fungal pathogen *C. albicans* has been used in several studies for the development of antibacterial materials in contact with the environment: dental implants, catheters, and contact lenses [29,53,63,64]. TiO_2_ coatings are effective in UV treatment [29], but even in dark conditions the adhesion of *C. albicans* is significantly reduced due to the increased hydrophilicity of the surface [53]. Another study tested a range of oxide ALD coatings (ZrO_2_, ZnO, HfO_2_, Al_2_O_3_, TiO_2_) using 16 strains of Gram-positive and Gram-negative bacteria [55]. However, only the disc diffusion method was used and the study lacks depth and has conflicting results.

### 3.2. Doped TiO_2_ and Combined TiO_2_

The ALD technique permits employing various reagents and their arbitrary mixture at diverse synthesis stages, making ALD auspicious not just for acquiring pristine oxide coatings but also for doped oxide systems. Therefore, incorporating reagents besides the titanium-containing precursor and oxidizing agent enables the doping of titanium oxide with diverse atoms [65]. Doping of titanium oxide can create further energy levels within its forbidden zone, generating a potent photocatalytic effect under both UV and visible radiation. This widens its potential usage as an antibacterial material given that only 4% of solar energy constitutes UV light [60]. The investigation of doped TiO_2_ via ALD has only been conducted in two works by a single scientific team [62,66]. The researchers studied thin films of TiO_2_, TiON, TiN, TiAlN, and TiO_2_:V_2_O_5_ tested with *E. coli* and *S. aureus*. All the coatings showed antibacterial activity, even in daylight, but the results were more effective when exposed to UV radiation. The doped films generally showed superior performance compared to pure TiO_2_ [62]. The authors investigated vanadium-doped TiO_2_ nanofilms on polypropylene (PP) hernia meshes in vitro and in vivo. The coatings were found to be highly effective in preventing the adherence of *S. aureus* and *E. coli*, and they showed excellent antibacterial activity. The research team of Takoudis et al. investigated mixed oxide systems consisting of TiO_2_ and ZrO_2_ on PMMA surfaces using *C. albicans* and *S. oralis* [63,64]. The TiO_2_-ZrO_2_ systems were found to be more effective than single oxides despite conflicting results on bacterial adhesion and survival.

In recent years, noble metal sensitization has emerged as an effective method for enhancing the photocatalytic and antibacterial activity of titanium oxide. This process reduces the recombination of photo-induced carriers at the noble metal/semiconductor interface [67]. Typically, noble metals such as silver [30,68,69] and gold [60] are deposited as nanoparticles on pre-synthesized ALD TiO_2_. While silver has been shown to possess powerful antibacterial properties due to Ag^+^ release, the formation of a Schottky heterojunction at the TiO_2_/Ag interface results in reduced electron–hole recombination rates. The consequence of this is the formation of an increased amount of bactericidal ROS when exposed to stimulated sunlight. Against *S. aureus* and *E. coli*, the antibacterial effectiveness of the TiO_2_/Ag coating was found to be 98.2% and 98.6%, respectively [30]. Notably, TiO_2_/Ag samples demonstrated distinct deformation and cracking in both strains. The authors suggest that the rupture of membranes is attributed to ROS. However, they also observed that cell walls of numerous bacteria were destroyed when silver nanoparticles (NPs) were used without TiO_2_. In the other study, the results of an almost 100% killing efficiency of the TiO_2_/Ag structures against *E.coli* was mainly achieved through Ag^+^ dissolution (Figure 6a–d) [68].

Furthermore, Au nanoparticles (NPs) can be used as efficient sensitizers. As such, tiny amounts of Au NPs that efficiently absorb visible radiation are ideal materials to enhance the photocatalytic properties of TiO_2_ on the surface of polyvinylidene fluoride (PVDF) membranes, exhibiting potent antibacterial activity towards *E. coli* under direct solar irradiation [60]. To understand such complex systems, it is necessary to continue to study and optimize properties such as the thickness and structure of the TiO_2_ layer, the size of the metal nanoparticles, and the density of the distribution on the surface. For example, according to research by Zhang et al., the efficacy against *E. coli* in dark and sunlight conditions is significantly increased as the particle size of Au decreases, resulting in an improvement in the antibacterial properties. In addition, the thickness of the TiO_2_ film has a significant effect on the hydrophilicity and efficiency of light radiation adsorption [60].

### 3.3. Zinc Oxide

#### 3.3.1. Mechanisms of ZnO Bactericidal Effect and Their Role

ZnO, as well as TiO_2_, is an n-type semiconductor with a band gap around 3.3 eV [70], meaning that absorption starts near the UV and it can generate oxidative ROS [71]. However, this is not the only factor that determines the antimicrobial activity of the coatings. Zn^2+^ ions can be released in the aqueous environment, easily transported across the cell membrane, interact with the active protease [72,73], and reduce intracellular adenosine triphosphate (ATP) levels in bacteria [74]. It has also been reported that nanoscale ZnO can bind to proteins or carbohydrates on the cell surface, modify the amino acid side chains, and cause cell death [75]. In addition, ZnO grows in the form of oriented crystallites, the morphology of which can lead to damage of the cell membrane during bacterial adhesion. Although the application of ZnO does not usually result in a significant increase in hydrophilicity, surface charge and surface energy factors may also be responsible to some extent for changes in the nature of bacterial adhesion. It is still debated which mechanism is most important and which is less important. To date, the influence of different mechanisms on the antibacterial properties of ALD ZnO has been investigated in several studies.

A number of studies show that the antibacterial activity of ZnO is either absent [54,70] or very low [76] under dark conditions (Figure 7). The importance of ROS is confirmed by data from Li et al. who investigated the effect of the active ROS scavenger vitamin E on reducing the antibacterial activity of 46 nm ZnO coatings against *E. coli* and *S. aureus* [77]. The question of exactly what kind of ROS are generated when ALD ZnO nanocoatings are exposed to UV radiation is still open. Park et al. recorded a significant amount of (a) ^•^O_2_^−^, (b) ^•^OH, and (c) ^1^O_2_ upon UV irradiation of 74 nm ZnO films on glass substrates [70]. The concentration of superoxide anions was found to be several orders of magnitude higher than that of the other ROS. Qian et al. showed that 33 nm coatings deposited on polypropylene (PP) nonwovens were active against *E. coli* even without light, but the activity increased significantly under xenon lamp illumination. In this case, the main ROS species that are produced under the illumination is the singlet oxygen [78]. However, even in the absence of illumination, small amounts of ROS can form due to defect structures. The oxygen vacancies present on the ZnO surface can interact with moisture to form superoxide radicals and H_2_O_2_ [35,79].

Despite this, several other studies of ALD ZnO have shown that the influence of UV exposure is minimal or absent and that the main factor in the antibacterial activity is zinc ions [58,80,81]. This is also supported by the fact that the deposition of thin layers of Al_2_O_3_, acting as a diffusion barrier on the surface of ZnO, reduces the activity against *E. coli* [82]. Also, the thicker the Al_2_O_3_ coating, the lower the antibacterial activity. This result directly indicates the importance of dissolution of zinc ions from the surface.

The difference in the above results can hardly be explained by the influence of the composition and structure of the obtained zinc oxide. The point is that in all the described works, diethyl zinc is used as an ALD precursor and, as a result, polycrystalline coatings with similar morphology are formed irrespective of the synthesis temperature. The coatings examined usually have a thickness of a few tens of nanometers, which is sufficient for significant ROS generation. The morphology of the substrate can have a small influence on the morphology of the ZnO, and indeed the substrates in the above-mentioned studies were different, but unfortunately no reliable conclusions about the influence of the substrate can be drawn from the data presented.

A detailed study of the influence of the contribution of different factors was carried out in the work where arrays of ZnO nanorods with different morphologies were coated with ALD Al_2_O_3_ [83]. It was shown that ZnO is effective even without light (96% inactivation of *E. coli*). At the same time, the application of a thin conformal layer of Al_2_O_3_ on the surface of ZnO reduces the antibacterial activity against *E. coli*, but it remains high (about 54%). This allowed us to conclude that the main factor in the activity is the relief and morphology of the surface, as ZnO nanorods can easily stab bacterial cells.

#### 3.3.2. The Effect of Thickness

As zinc oxide is relatively soluble, the thickness of the coatings plays an important role in the duration of the antibacterial effect. However, excessively thick coatings can have adverse effects on certain applications, such as medical implants. For example, Wu et al. showed that relatively thick ZnO films (more than 50 cycles) on mesoporous TiO_2_ released too much zinc, which negatively affected the viability, proliferation, and differentiation of MC3T3-E1 osteoblasts [84]. They also showed that the mesoporous structures overgrow with relatively thick coatings and cannot fulfil the role of drug carriers. A negative effect on MG-63 cell proliferation of 195–271 nm thick films was reported [26]. Cytotoxicity of 26 and 52 nm films on HS-27 fibroblasts was also observed [85]. Our recent work also showed the negative effect of 40 nm ZnO on the viability of MG-63 pre-osteoblasts and mesenchymal stem cells FetMSCs [61]. Li et al. found that 76 nm coatings did not significantly affect the adhesion and viability of MC3T3-E1, but they had a negative effect on spreading [86]. However, in another study, it was shown that very thin films around 5.3 nm conversely improved the adhesion and spreading of MC3T3-E1 [87]. Indeed, research into the biocompatibility of ZnO coatings with varying thicknesses of less than 10 nm using MC3T3-E1 revealed a two-fold impact on the cytological response. While cell differentiation significantly increased, cell proliferation decreased with increased coating thickness [43,84].

A similar situation is observed with bacterial cells. Puvvada et al. showed that ultra-thin zinc oxide films are not only ineffective, they actually promote bacterial growth, but from a thickness of about 2 to 10 nm (10–50 ALD cycles), the antibacterial effect begins to manifest itself sharply [36]. A similar threshold of several nanometers (30 ALD cycles) for *E. coli* and *S. aureus* has been reported in another study [43] (Figure 8). At lower thicknesses, the opposite effect is observed and the authors suggest that Zn^2+^ at low concentrations acts as a nutrient for bacterial growth. In general, the vast majority of studies with films thicker than 5–10 nm confirms that increasing the thickness significantly improves the antibacterial effect [34,35,54,58,79,84].

However, there are several studies that found no effect of ZnO thickness on *E. coli* and *S. aureus*. For example, no difference in antibacterial activity was found between 26 and 52 nm ALD ZnO according to [85]. Kääriäinen et al. found no difference in antibacterial properties for 45, 95, and 285 nm thick films [71]. And Basiaga et al. states that no significant differences were found in the CFU adhering to the surface of 316LVM steel depending on the number (500 and 1500) of ALD cycles of ZnO [26]. It can therefore be concluded that a thickness of a few tens of nanometers is sufficient to achieve a significant antibacterial effect. The effect is observed even under dark conditions. Therefore, the elution of Zn^2+^ is probably responsible for the antibacterial properties of ZnO films.

As zinc oxide is soluble in an aqueous medium, it is crucial to focus on the changes in antibacterial properties that may occur. Lee et al. showed that the significant antibacterial activity of 55 nm coatings against *E. coli* was observed within the first few tens of minutes and reached a maximum after 1 h [54]; however, saturation of zinc dissolution was observed within the first few days for 100 nm coatings [80]. According to Puvvada et al., the antibacterial effect increases with time and reaches a maximum at the end of 1 day [36]. Considering that the ALD coatings are very thin, it is important to know how long the antibacterial effect lasts. Yao et al. showed that thin coatings of about 5 nm on flat polished Y-ZrO_2_ are effective in the first days of presoaking, and the activity disappears on days 7 and 14 [87]. At the same time, it remains on the sand-blasted and acid-etched ZrO_2_ (SLA-ZrO_2_) surface with an analogue coating. It can therefore be concluded that coatings of a few nanometers are sufficient for use in medical implants, as significant antibacterial activity is only required in the initial few days after surgery, whereas thicker coatings are required, for example, for water purification or antibacterial textiles.

#### 3.3.3. The Effect of the Deposition Temperature

Several studies have investigated the effect of ALD temperature on the functional properties of ZnO coatings. According to the results of dynamic tests, Lee et al. showed that under UV exposure, the number of surviving *E. coli* decreases significantly in the transition from weakly crystalline ZnO obtained at 70 °C to fairly well crystallized ZnO obtained at 100 °C [54]. Increasing the synthesis temperature further to 125 and 200 °C slightly increases the number of surviving bacteria. Increased survival and improved adhesion of *E. coli* and *S. aureus* when the ALD temperature was increased from 100 to 300 °C has also been reported in [26]. Wang et al. showed that when ZnO was post-annealed between 225 and 275 °C, the change in survival of oral bacteria, including *S. sanguinis*, *Bifidobacterium* and *Porphyromonas gingivalis*, was non-monotonic and decreased between 200 and 250 °C, but it increased at 275 °C [80]. This characteristic of the effect of annealing temperature on survival correlates very well with the release of Zn^2+^. Thus, although the synthesis temperature of ZnO has almost no effect on the structural and morphological characteristics, there is an influence on the antibacterial properties, but it is still poorly understood.

#### 3.3.4. The Effect of the Morphology, Topography, and Wettability

Surface morphology, topography, surface free energy, surface charge, and wettability are also important surface properties that influence antimicrobial activity. Furthermore, these characteristics are interrelated and must be considered together. Since the ALD process produces conformal and homogeneous coatings, the resulting topography and morphology are usually determined by the morphology and topography of the substrate. For example, Basiaga et al. showed that the polished surface of 316LVM steel with ZnO coatings was much less favorable for the adhesion of *E. coli* and *S. aureus* than the micro-rough surface obtained via SLA technology and subsequent ALD of ZnO [26]. Indeed, surface inhomogeneities comparable to the size of the bacteria can favor their adhesion (Figure 4b). However, nanoscale roughness should enhance the bactericidal properties, since in this case the specific surface area increases significantly and the release of Zn ions increases accordingly [80]. Experimental data show that even micro-roughened ZrO_2_ substrates obtained by SLA become extremely active against *E. coli*, *S. aureus*, and *P. gingivalis* after ZnO deposition [87]. In addition, it has been shown that nanostructures with sharp edges or rods can pierce the cell membrane and lead to bacteria death [83] (Figure 9). Although ALD produces uniform films, their low thickness means that the influence of the substrate affects the wettability and surface energy. As a result, depending on the substrate type, ZnO films can exhibit both hydrophilic properties when grown on inorganic substrates [79,80,84] and hydrophobic properties when grown on polymeric substrates [43,77,78].

#### 3.3.5. Effects on Various Types of Bacteria and Viruses

The issue of ZnO’s antibacterial activity against different strains is very interesting. ZnO is very effective against the Gram-negative *E. coli* and the Gram-positive *S. aureus* strains. A comparative analysis of the results shows that in most cases the antibacterial activity against *S. aureus* and *E. coli* is either close [84,85,88] or higher for *S. aureus* [26,34,43,58,78]. However, in the dynamic cases, the antibacterial activity against *E. coli* was maintained for several hours, while the antibacterial activity against *S. aureus* decreased [77]. The authors explain this by the greater sensitivity of the *E. coli* membranes to ROS. Similar results were obtained for cellulose fibers coated with Al_2_O_3_/ZnO bilayer nanofilms [76]. *E. coli* were completely inactivated after only 30 min of UV irradiation, whereas *S. aureus* required at least 45 min of light exposure for 100% inactivation.

Of particular interest is the study where ZnO coatings were prepared with a surface enriched with either zinc or oxygen [79]. The oxygen-enriched surface was found to release more Zn^2+^ and ROS, resulting in greater efficacy against *S. aureus*. At the same time, the zinc-enriched surface releases more H_2_O_2_, which is more active against *E. coli* (Figure 10).

Despite the apparent efficacy against *E. coli* and *S. aureus*, ALD ZnO is not effective against all strains. For example, Skoog et al. showed a lack of activity against *Pseudomonas aeruginosa*, *Enterococcus faecalis*, and *Candida albicans* [89]. Vaha-Nisi et al. reported a lack of activity against the bacterium *Bacillus subtilis* and a fungus *Aspergillus niger* [82].

Antiviral activity has also been investigated in several papers, but the results are rather contradictory. Kumar et al. showed the efficacy of ZnO applied to fibrous silk against the coronavirus and rhinovirus [35], and Dicastillo et al., despite high antibacterial activity against *E. coli* and *S. aureus*, found no activity against the norovirus [90].

### 3.4. ZnO-Based Nanocomposites and Nanolaminates

As in the case of titanium oxide, noble metal sensitization can also be effective for zinc oxide. In the work of Di Mauro et al., ALD Ag NPs with sizes ranging from 4 to 25 were synthesized on a previously prepared ALD ZnO ultrathin film (Figure 11a–d) [38]. Such heterogeneous photocatalysis showed high efficiency in the decomposition of various organic pollutants in aqueous solution, methylene blue (MB), paracetamol, and sodium lauryl sulfate, but the authors paid very little attention to antibacterial properties. The obtained ZnO-Ag systems were very effective against *E. coli* in dark and UV light conditions (Figure 11f), but unfortunately the authors did not consider the effect of silver particle size on the bactericidal properties. Akyildiz et al. synthesized photocatalytic systems by depositing a 20 nm layer of ZnO on cotton fibers followed by photochemical deposition of Ag NPs [33]. However, the antibacterial effect of silver was too high to draw any conclusions about metal sensitization of ZnO. ZnO nanorods decorated with Ag NPs were very effective against *E. coli* according to another study [91].

Liu et al. synthesized cubic Cu_2_O particles on the surface conductive and transparent indium tin oxide (ITO) glass [31]. Then, ZnO coatings were applied for the formation of the p-n heterojunction which limit recombination of electrons and holes and improve photocatalytic efficiency. This heterojunction was effective against *S. aureus* both at dark and simulated sunlight conditions (Figure 12). The results showed that the bactericidal effect was achieved due to both a high effective ROS formation and Cu/Zn ions elution. Sr doped titania nanotubes were coated with 30 cycles of ZnO and then grafted with octadecylphosphonic acid (OPDA) [44]. Sr-ZnO composites showed high activity against *S. aureus* and *E. coli*, but the OPDA coating decreased activity due to shielding and reducing the dissolution of Zn ions.

Recently, several mixed zinc oxide-based oxide systems have been successfully obtained. Li et al. synthesized ZnO/FeOx nanolaminates using 200 cycles of ALD (21–25 nm thickness) consisting of two, twenty, and one hundred alternating layers (Figure 13a) [92]. The nanolaminates were activated using simulated solar light and light-emitting diode (LED) light. The authors believe that such nanolaminates facilitate the transfer of numerous photoelectrons from ZnO to Fe_2_O_3_ under light irradiation, thus generating a significant amount of ROS. The best antibacterial activity against *E. coli* and *S. aureus* was observed for the twenty-layer sample in both light regimes (Figure 13b–e). At the same time, the nanolaminates showed high antiviral activity against H1N1 (Figure 13f–m) and no cytotoxicity against NIH-3T3 fibroblasts. The obtained heterostructures retain high transparency, have high abrasion resistance, and can be used as effective antibacterial coatings for touchscreens. ALD coatings based on ZnO and Al_2_O_3_ (AZO), obtained in different ratios but with the same thickness of 50 nm, were investigated for antibacterial and antifungal properties [32]. AZO showed high efficacy against bacteria (*E. coli* and *S. aureus*) and five commonly found in real-life molds, both in the presence of natural light and in the absence of light. The Zn ion elution of AZO was significantly lower than that of pure zinc oxide, which is very important for the development of antibacterial moisture barrier films for next generation flexible touch display functional optical films. 

Recently, we have investigated complex oxide systems based on ZnO and TiO_2_ (ZTO) with a thickness of about 40 nm [61]. ZnO and ZTO samples were very effective against *S. aureus*, *A. baumannii*, and *P. aeruginosa*. In this case, unlike pure zinc oxide, ZTO had no negative effect on the adhesion of fetal mesenchymal stem cells (FetMSCs).

### 3.5. Other Binary and More Complex Oxide Compounds

Many other oxide compounds can be produced through ALD, in addition to titanium oxide and zinc oxide. It is possible that some of these compounds also possess antibacterial properties. Godlewski et al. compared a series of rather thick films (about 200 nm) of wide bandgap oxides (ZnO, HfO_2_, TiO_2_, ZrO_2_, Al_2_O_3_) deposited on the surface of paper discs [55]. The authors tested 16 Gram-positive and Gram-negative strains using the agar disc diffusion method without illumination. No bactericidal effect was found for *B. cereus*, *S. agalactiae*, *Streptococcus alpha-hemolytic*, *E. coli*, *Klebsiella* spp., *Salmonella BO*, or *Yersinia* spp., but an effect was found against *B. subtilis*, *S. aureus* ATCC and MRSA, *S. epidermidis*, *S. pseudintermedius*, *Streptococcus beta-hemolytic*, *Proteus* spp., and *Salmonella DO;* most oxides were active. Compared to ZnO, ZrO_2_ and HfO_2_ nanocoatings showed similar or better antimicrobial activity [55]. ZrO_2_ is a biocompatible material that stimulates stromal and mesenchymal cell proliferation and differentiation in the osteogenic direction [93,94,95], but 20 nm ZrO_2_ coatings reduce the adhesion of *S. mutans* and *P. gingivalis* [96]. Two recent studies have demonstrated the high efficiency of TiO_2_-ZrO_2_ complex oxide systems [63,64] against *C. albicans* and *S. oralis* on the PMMA surface. No significant antibacterial activity was observed for pure ZrO_2_.

It is noteworthy that according to Godlewski et al., Al_2_O_3_ performed even better than TiO_2_ in most of the cases [55]. As shown in [85], Al_2_O_3_ coatings of 13.5 and 27 nm thickness on cellulose-based fibers can also exert an antibacterial effect against *S. aureus* and *E. coli*, which is slightly lower than that of ZnO coatings. At the same time, Al_2_O_3_, unlike ZnO, was shown in the same study not to be cytotoxic against the human HS-27 fibroblast and human keratinocyte standardized cell line (HaCaT). The antibacterial effect of 12.5 and 25 nm thick Al_2_O_3_ coatings against *E. coli* has also been demonstrated [82]. In general, all authors agree that Al_2_O_3_ is less active than ZnO [55,82,85].

Of particular interest are pair of papers by Li et al. [86,97]. The authors prepared two-dimensional porphyrinic zinc and copper containing a metal−organic framework (2D MOF) coated with Fe_2_O_3_ using the ALD. Such materials have been shown to be effective in vitro and in vivo against various oral pathogens (*P. gingivalis*, *Fusobacterium nucleatum*, and *S. aureus*) due to both ROS formation and released ions (Figure 14). The enhanced photocatalytic activity of the ALD modified MOFs results from the lower adsorption energy and more charge transfer due to the synergistic effect of metal-linker bridging units, abundant active sites, and the excellent light-harvesting network. The in vivo results showed that these FeOx-modified MOFs can be used in photodynamic ion therapy of periodontitis with a greater therapeutic effect than the reported clinical treatment with minocycline and vancomycin. In another study, it was shown that visible light transparent ALD ZnO–Fe_2_O_3_ superlattice nanocoatings are effective bactericidal and virucidal materials (H1N1) [92]. The action of such superlattices is based on the photocatalytic mechanism, where ZnO and Fe_2_O_3_ served as electronic donors and receptors, respectively. The heterointerface between ZnO and Fe_2_O_3_ facilitated the transfer of numerous photoelectrons from ZnO to Fe_2_O_3_ under light irradiation and promoted the ROS formation.

Furthermore, in addition to the aforementioned familiar compounds, obscure substances have exhibited antibacterial properties. For example, Radtke et al. demonstrated significant inhibition in the staphylococcal colonization and biofilm formation for titanium oxide nanotubes (TNT) with ALD-coated CaCO_3_ subsequently converted to hydroxyapatite [98]. In previous work by this research group, TNT formed on the Ti6Al4V alloy were shown to have in vitro antibiofilm properties themselves when tested on the *S. aureus* model [99]. But the double modified biomaterials exhibited stronger antimicrobial activity than the surface with TNT, indicating a synergistic effect of both modifications [98]. A recent paper by Saha et al. demonstrated the antibacterial activity of conformal MgO nanocoatings deposited on collagen membranes against polymicrobial biofilm from human saliva [64].

TaN is widely used for the implant coating material due to its high mechanical strength, high chemical stability, and high biocompatibility but TaN coating also has a tremendous antibacterial effect [28]. TaN coating decreased *E. coli* and *S. aureus* colony formation due to the increased water contact angle and hydrophobic properties.

### 3.6. Silver and Other Metals

Metallic silver is one of the best known and most successful antimicrobial, antifungal, antiviral, and anti-inflammatory materials [100]. Silver has been used as an antibacterial and medicinal agent for thousands of years [101], but its importance declined with the discovery of antibiotics. However, the problem of the emergence of antibiotic-resistant strains soon arose. This increase in antibiotic resistance has led to renewed interest in the use of silver as an antibacterial agent [102].

The antimicrobial activity of Ag-coated surfaces is commonly attributed to the strong oxidative effect of silver in the course of the gradual release of silver ions into the biological environment. However, the possible mechanisms of silver’s antibacterial action are quite diverse. Ag ions induce the oxidative stress effect on cell membranes, damaging and penetrating them. In addition to its own chemical activity, silver ions can induce the formation of reactive oxygen species (ROS), which can also damage organelles and impair their function. Ag ions have many possible ways to disrupt the functioning of cell organelles and biochemical processes, such as respiratory activity, intracellular ATP levels, and inhibition of the DNA replication. This is due to the fact that silver ions are Lewis acids and can easily interact with thiol and amino groups of proteins, nucleic acids, and others [101]. Such diverse and potent effects make silver effective against Gram-positive and Gram-negative bacteria. However, this high activity can result in a negative cytotoxic response of body cells and cause silver accumulation in the internal organs.

However, despite these problems, a number of studies have shown that ultrathin nanosilver coatings do not exhibit this cytotoxicity. The effect of silver on different cells and tissues depends on both the size of the Ag structures and the cell type being studied in vitro. Several studies have shown that nanoparticles of 50 nm or less at low concentrations are not cytotoxic to mouse embryonic stem cells (mESC) [103], human umbilical vein endothelial cells (HUVEC) [104], normal human fibroblasts (FN1) [104], human mesenchymal stem cells hMSC [105,106,107], L929 fibroblasts [108,109], and human keratinocyte cell line HaCaT [110]. However, at high concentrations, such nanoparticles are toxic, as has been demonstrated in human embryonic kidney HEK293 [111,112], mouse embryonic fibroblast cells NIH3T3 [113], and human dermal fibroblasts HDFs [114].

The importance of dose and AgNPs size for their toxicity has also been demonstrated in a number of in vivo studies using different routes of administration (inhalation, intravenous, and intraperitoneal) [115,116,117,118,119]. Thus, the issue of nanoparticle size and concentration, and, in the case of nanofilms, their thickness and morphology, is the most important issue in studying the medical applications of silver.

#### 3.6.1. ALD of Silver Nanoparticles and Nanocoatings

Although ALD is very good at producing oxides and a number of other binary compounds, there are approaches that allow the production of metallic films. However, continuous metallic nanofilms are not easily obtained via ALD. The problem is that the growth of metals suffers from poor nucleation and initiates island growth [120,121]. This is particularly characteristic of silver ALD, where NPs grow in a relatively small number of cycles (Figure 15). Many ALD cycles are therefore required to obtain a continuous film. Nevertheless, this feature can be considered positive, as nanoparticles increase the specific surface area and hence the dissolution rate of silver ions compared to continuous films.

Although silver has been produced via ALD for a very long time, there are only a few papers investigating the antibacterial properties of ALD-Ag (Table 2). Radtke et al. investigated silver NPs with diameters of 7.8–9.2 nm deposited on titanium oxide nanotubes with diameters ranging from 10 to 84 nm [102]. An inhibitory effect on the biofilm formation of *S. aureus* was found using the suspension method of study. The samples with the smallest diameter of nanotubes were the most effective as they have a large specific surface area.

Recently we synthesized larger diameter silver NPs (16–30 nm) on the surface of flat titanium discs. The results showed the inhibitory effect of silver on the growth of *S. aureus.* The ALD synthesis of an additional sublayer of crystalline TiO_2_ enhanced the inhibitory effect [57]. A similar synergistic effect on *E. coli* was observed in [68], where the antibacterial properties of an ALD layer of TiO_2_ and Ag NPs prepared via photodeposition were investigated. This effect has been discussed in more detail in Section 3.2.

Several studies have shown that, in addition to their antibacterial effect, ALD silver NPs can have a positive effect on the adhesion and proliferation of fibroblast and osteoblast-like cells [57,102,122]. In vivo studies of Ag-decorated porous titanium scaffolds in tibial defects in rats showed that Ag NPs induced robust osteogenesis and angiogenesis with no evidence of systemic toxicity and silver accumulation in the liver [27,122]. Further in vivo studies by the authors over 12 weeks showed that Ag accumulation occurs within the osseous tissue immediately adjacent to the implant surface. TEM and selected area electron diffraction (SAED) show that silver is part of the less toxic Ag_2_S within the newly formed bone. (Figure 16) [27].

In terms of practical applications, the work [123], in which Ag nanoislands were obtained on the surface of N95 medical masks at relatively low temperatures (90 and 120 °C) via ALD is of great importance. Such coatings can be very effective in increasing the service life and reliability of personal protective equipment (PPE), as such particles prevent microbial contamination and accumulation in the membranes and fibers of PPE. In addition, the prepared Ag nanoparticles were stable to washing. According to the authors, ALD is ideal for coating PPE because it provides high adhesion of nanoparticles through chemisorption, does not require the use of capping agents, is performed in the gas phase, which can be important for liquid-sensitive materials, and avoids blocking the pores and negatively affecting the air permeability of the mask.

Although the results [123] show an improvement in the suppression of biofilm formation with smaller silver particles, the effect of particle size on its antibacterial activity and cytotoxicity/biocompatibility remains controversial. The duration of the silver effect and the onset of action also remain open, as the results showed that 2 h was not sufficient for effective bacteria/Ag contact, whereas after a longer time (24 h) effective contact occurred, resulting in a reduction in biofilm formation [123].

A major problem in silver ALD is the lack of a good precursor. A reproducible and controlled ALD process requires sufficient vapor pressure and thermal stability of the precursor. The most successful precursor is (2,2-dimethyl-6,6,7,7,8,8,8-heptafluorooctane-3,5-dionato) silver(I) triethylphosphine (Ag(fod)(PEt_3_)). However, to achieve the necessary vapor pressure, it must be heated to temperatures above 80–100 °C, which can lead to the slow decomposition of Ag(fod)(PEt_3_). Furthermore, the optimum growth temperature for this compound varies within a fairly narrow range of 120–160 °C [24].

In addition, most synthesis techniques for Ag production via ALD use hydrogen plasma as a reducing agent. As has been shown in a number of papers, the use of plasma as a co-reagent causes problems in the treatment of complex-shaped surfaces [124]. Plasma is very unstable and quickly “loses its activity” when treating deep channels. Therefore, it is necessary to carry out studies on the selection of an optimal reducing agent. In [123], borane dimethylamine complex ((CH_3_)_2_NH*BH_3_) was tested as a co-reagent. The results indeed showed silver growth in the inner layers of the fibrous structure of the N95 filter media.

#### 3.6.2. ALD of Other Metallic Nanocoatings

Besides silver, several metals including copper, gold, and iron have strong antibacterial properties [125,126]. These metal layers have been successfully fabricated via ALD, but only two studies have investigated their antibacterial properties. TiO_2_ inverse opal structures were functionalized with an ultrathin ALD Cu layer (1.1 nm Cu loading) [127]. A significant reduction in the CFU count of *S. aureus* was observed on the Cu containing samples pretreated at 550 °C to increase the stability of the Cu particles.

Yang et al. conducted an antimicrobial study of a whole series of nanozymes based on ALD-Pt ultra-small nanoparticles (0.55–2.81 nm) prepared on the surface of carbon nanotubes (CNTs) (Figure 17) [128]. The nanozymes with a Pt NPs size of 1.69 nm exhibited much higher antibacterial activity than the others in the presence of H_2_O_2_ due to the effective formation of toxic free radicals. In the absence of H_2_O_2_, CNTs/Pt NPs showed a negligible killing effect against both *E. coli* and *S. aureus*.

## 4. Applications for ALD Antibacterial Coatings

In most of the studies analyzed in this review, the authors focused on specific applications of the antibacterial coatings produced. Therefore, the antibacterial properties were analyzed using different specific bacterial strains and other functional properties of the materials and coatings were investigated. For implants, in addition to antibacterial properties, cytotoxicity was evaluated for different host cells such as pre-osteoblasts, fibroblasts, and epithelial cells, and an appropriate set of strains was selected depending on whether the implant was internal or in contact with the environment. For PPE and air purifying coatings, air permeability was studied and for water purifying coatings, stability in aqueous media was studied.

### 4.1. Medical Implants

The development of effective antibacterial coatings for orthopedic implants is no less important than the development of coatings to accelerate their biointegration. The majority of orthopedic implant rejections and failures are due to the formation of a biofilm of MRSA or *S. epidermidis* that prevents drug penetration to the infected site [122]. Such problems most often result in the need for costly and often painful reimplantation, and, in some cases, post-infection morbidity and mortality. It is therefore difficult to solve this problem by using traditional clinical antimicrobial therapy methods, such as oral administration or intravenous injection [129,130]. Therefore, coatings that provide sustained release of antibacterial agents and inhibit biofilm formation are of great practical importance. Coatings that release metal ions (e.g., Zn^2+^, Ag^+^, and Cu^2+^) are very effective as they have broad-spectrum antibacterial properties and are effective even against antibiotic-resistant strains [131,132].

To date, there are about ten studies that have focused on antibacterial ALD coatings for potential applications in orthopedic implants, and there have been a slightly smaller number for dental implants. Substrates tested have included flat titanium [28,49,57,96], ZrO_2_ [80,87], and stainless steel [26], as well as substrates with deposited carbon nanotubes [43], titanium nanotubes [44,98], or rough SLA surfaces [26,87]. In addition, porous substrates, 3D-printed scaffolds made of titanium [27,122] and iron [86], as well as the denture material PMMA [53,63,64], have been used. Of particular note is the study [133] where osteopromotive MgO coatings were applied to the surface of collagen, which is an integral part of bone, cartilage, skin, and tendon tissue. ZnO [26,43,44,80,86,87], TiO_2_ [53,57,61], MgO [133], hydroxyapatite [98], ZrO_2_ [96], TaN [28], Ag [27,57,122], and TiO_2_-ZrO_2_ complex oxides [39,40] have been most commonly used as coatings [63,64].

In the majority of cases for orthopedic implants, antibacterial properties were evaluated using *S. aureus* and *E. coli*, and, less frequently, MRSA [49,122] and *S. epidermidis* [122]. For dental implants, the set of strains included *Porphyromonas gingivalis* [80,87,96], *S. Sanguinis*, Bifidobacterium [80], *S. oralis* [63], *C. albicans* [53,63,64], and antibacterial polymicrobial biofilm from human saliva [133]. In addition to the antibacterial properties, in vitro biocompatibility studies were performed using osteoblast-like MC3T3-E1 [43,44,84,86,87,96,134], MG-63 [26,57], and SAOS-2 osteosarcoma [122], fibroblasts (L929) [26,98,134], and gingival keratinocytes as well as epithelial cells [80,81,122].

In the absolute majority of studies, the coatings showed high antibacterial efficacy against both standard *S. aureus* and *E. coli*, as well as against specific strains, but it is difficult to compare them, as the studied samples differ greatly in thickness, morphology, used substrates, and measurement techniques. Nevertheless, it can be stated that zinc oxide and silver coatings have the strongest antibacterial properties. At the same time, biocompatibility studies have been ambiguous for zinc oxide coatings. As demonstrated earlier in the text, most researchers have found that ALD ZnO does not pose a threat to eukaryotic cells. Nevertheless, our analysis revealed detrimental results of 40 nm thick zinc oxide on FetMSCs [61]. Basiaga et al. found a negative effect of thicker coatings on MG-63 proliferation [26], and Zhu et al. found a decrease in the viability of MC3T3-E1 with increasing coating thickness. In contrast, silver nanoparticles obtained in three studies showed a rather high biocompatibility and promotive effect on the differentiation of osteoblast-like MG-63 and FetMSCs in vitro [57] and no negative effect on new bone formation, robust osteogenesis, and angiogenesis in in vivo [27,122]. In addition to the above, only a few studies have used the in vivo approach to evaluate the prospects of ALD coatings. The similar Wistar rat (*Rattus norvegicus*) in vivo model was used to confirm the biocompatibility of ALD MgO nanocoatings and to increase angiogenesis and bone formation [133]. The thicker coatings (500 vs. 200 cycles) were better candidates for guided bone regeneration due to the higher release of magnesium ions.

Xue et al. investigated 3D customized earplugs coated with ZnO nanoaggregates produced by combined ALD and hydrothermal methods [134]. The in vitro study showed that the coatings were highly effective in preventing the growth of five major pathogens from patients with otitis media. A study of earplugs pre-impregnated with MRSA solution in an in vivo rat model showed that the subcutaneous tissue was more infected in the absence of ZnO coatings.

In addition to antibacterial effects and biocompatibility regulation, ALD coatings, due to their high quality and homogeneity (Figure 1), can fulfil the role of a reliable biocorrosion rate regulator for biodegradable implants and scaffolds, which are widely used for bone repair and regeneration, but they are also used for vascular stents. For example, Li et al. showed that 76 nm thick ZnO coatings not only have strong antibacterial activity against both *E. coli* and *S. aureus*, they do not significantly affect cytocompatibility with MC3T3-E1 cells, but they also significantly slow down biocorrosion of 3D porous iron scaffolds [135].

### 4.2. Antibacterial Textile and Personal Protective Equipment

The COVID-19 pandemic has increased interest in the development of antibacterial textiles and effective PPE, and as a result the number of studies on ALD coatings for these applications has become very high in recent years. Both natural materials (cotton, silk) [33,34,36,76,85] and man-made polyamide [58,136], viscose [85], polypropylene [78], and combinations of natural materials and polymers [35,77] are used as substrates for coating. In the case of cotton and silk, the problem of bacterial contamination is particularly acute. These materials absorb large amounts of water and the aqueous environment is very conducive to the growth of bacteria [34].

In almost all the studies, zinc oxide is used as the coating material, which is significantly superior to ALD TiO_2_ deposited on polyamide 66 [58] and Al_2_O_3_ deposited on the surface of cotton and viscose [85]. Only in the study by Astaneh et al. did the authors choose silver nanoparticles as the antibacterial material deposited on the surface of N95 masks (the results are described in more detail in Section 3.6.1). Composites of ALD ZnO and photodeposited Ag nanoparticles deposited on the cotton surface were very effective against *S. aureus* and *E. coli* under the conditions of light irradiation and dark [33].

The thickness of the deposited coatings varied from fractions of 1 nm [15] to 116 nm [34]. Even relatively thick (>100 nm) coatings had little effect on the air permeability of the material [34], which is particularly important for PPE and textile applications.

Almost all studies to date have shown some antibacterial activity of ZnO in both light and dark experiments using *E. coli* and *S. aureus*. For relatively thin films (up to 45 nm), an increase in thickness has a positive effect on the antibacterial activity [35]. In addition to the antibacterial properties, Kumar et al. investigated the antiviral activity [35]. ZnO 45 nm coatings deposited on electrospun nanofibrous silk showed a 95% reduction in infectivity for coronavirus (OC43: enveloped) and rhinovirus (RV14: non-enveloped) after 1 h of white light illumination and a 99% reduction for RV14 after 2 h of illumination. It is worth noting that ZnO coatings with thicknesses of approximately 25 and 50 nm showed little cytotoxicity towards *human fibroblasts* and *keratinocytes* in contrast to similar Al_2_O_3_ coatings [85], which is particularly important for textile and PPE applications.

### 4.3. Purification and Disinfection of Water and Air

The chemical disinfection of water is a very effective, simple, and relatively inexpensive method. However, chemical reactions produce a number of toxic by-products that are not only health-threatening but also cause the water to taste bad. Therefore, photocatalytic ROS-based methods have attracted increasing interest. It is important to note that photocatalytic ROS generation not only kills bacteria but also destroys many dangerous organic molecules and contaminants [37]. So far, the functional properties of both the pure oxide photocatalysts, ZnO [38] and TiO_2_ [39], and the composite materials ZnO-Ag [38], TiO_2_-Au [60], and TiO_2_-Ag [68], have been examined. Polymeric porous membranes based on PVDF, PTFE, and PMMA were commonly used as substrates. Pure oxide systems are less effective than metal oxide composites both as antibacterial coatings and as photocatalysts to degrade various organic contaminants such as MB, parecetomol, sodium lauryl sulfate, and methyl orange [38,60]. Zhang et al. note that although TiO_2_ nanoarrays prepared via ALD and solvothermal methods reduce water flux from 350 to 200 L/m^2^∙h with long-term use, the reduction is compensated by the absence of biofilm overgrowth in the pores of the PTFE ultrafiltration membrane [39]. In addition to solving membrane biofilm overgrowth, it should be noted that ALD antibacterial coatings have the important property of high adhesion, which is very important for the long-term use of ultrafiltration membranes.

The use of ALD antimicrobial coatings also provides an effective solution to the problem of air purification. Zhong et al. obtained coatings of ALD ZnO seed layers and hydrothermally grown ZnO nanorods on the surface fiber matrix of ePTFE filters (Figure 18) [88]. These filters have a high antibacterial activity (>99.0% sterilization rates against *S. aureus* and *E. coli*) and are 40% more efficient in the capture of 0.30 and 1.2 μm simulated dust particles. Although the densely packed ZnO nanorods significantly altered the pore structure of the original expanded polytetrafluoroethylene (ePTFE) filter, the change in flow resistance was negligible. The application of silver nanoparticles to the ZnO nanorods resulted in even higher antibacterial efficacy (>99.9%) [91].

### 4.4. Other Application

The problem of disinfecting various high-frequency touch surfaces can be effectively solved with stable antibacterial coatings. ALD is a very successful method for the synthesis of ultrafine coatings with a thickness of no more than a few tens of nanometers. Such coatings have high transparency and can be used for touch screens with permanent antibacterial properties. In recent years, work has been carried out on the synthesis of complex systems, TiO_2_-Ag [30], Cu_2_O-ZnO [31], ZnO-Fe_2_O_3_ [92], and Al_2_O_3_-ZnO (AZO) [32], on transparent substrates such as glass (ITO) and polyethylene terephthalate (PET). The thicknesses of the coatings varied in the range of 10–50 nm and the coatings showed high transparency in the visible range. At the same time, regardless of the composition, they showed high activity not only against *E. coli*, *S. aureus* [30,31,32,92], MRD, and MRSA [92] but also against several fungi such as *Aspergillus brasiliensis*, *Talaromyces pinophilus*, *Chaetomium globosum*, *Trichoderma virens*, and *Aureobasidium pullulans* [32]. All researchers agree that external illumination, at least sunlight, is required for the greatest efficiency of coatings, but Li et al. showed that LED illumination can be a substitute for sunlight and UV for photocatalytic disinfection using ZnO-Fe_2_O_3_ nanolaminates, which is relevant to modern personal electronic devices with LED backlit screens.

Transparent coatings may also be important for food packaging, although such an application is only discussed in one publication [82], where ZnO and Al_2_O_3_ thin films were deposited on biaxially oriented polylactic acid (BOPLA) polymer films that were 40 μm thick and 50 μm thick top-side with corona-treated polypropylene. Both types of coatings showed good oxygen and vapor barrier properties, but ZnO was more active against *E. coli*, and some of the ZnO samples were effective against *Bacillus subtilis* and the fungus *Aspergillus niger*.

To date, there have been several studies that have tested antimicrobial ALD coatings for potential use in treating various diseases and injuries. The idea of using alumina membranes conformally coated with 8 nm ZnO for the treatment of burns and other skin wounds was proposed by Skoog et al. in 2012 [89]. In vitro studies with specific strains showed that the materials were effective against *B. subtilis*, *E. coli*, *S. aureus*, and *S. epidermidis*, but they were completely ineffective against *P. aeruginosa*, *E. faecalis*, and *C. albicans*. Recently, Li et al. found high photocatalytic activity in metal-organic frameworks (MOFs) coated with ultrathin Fe_2_O_3_ layer (20 ALD cycles) [86]. *S. aureus*, *E. coli*, and an in vivo cutaneous MRSA wound infection model were used to validate the prospective application of such composites. At 2 days post-infection, the wound in the MOF-Fe_2_O_3_ group showed excellent MRSA killing (99.9995 ± 0.0003%), no obvious inflammatory response, and complete escharosis in vivo. The group that did not use MOF-Fe_2_O_3_ treatment showed severe suppuration due to heavy MRSA loads in the wounds [86]. Another study by this research group showed high antibacterial activity of MOF-Fe_2_O_3_ against oral pathogens (*Porphyromonas gingivalis*, *Fusobacterium nucleatum*, and *S. aureus*). The authors believe that the obtained heterostructures can be effectively used for photodynamic ion therapy of periodontitis due to the rapid antibacterial activity, alleviated inflammation, and improved angiogenesis [97].

In addition to the above-mentioned directions, the work of Yang et al. is of interest as they proposed using a composite of carbon nanotubes and ultra-small ALD platinum particles to obtain enzyme-like materials—nanozymes [128]. Also of great interest are the studies of TiO_2_-coated alumina membranes [51] and mesoporous TiO_2_/ZnO composites [84], which the authors suggest could potentially be used for efficient drug loading.

## 5. Analysis of the Results

### 5.1. Comparison of Different Antibacterial Coatings

To date, a variety of oxide and metallic ALD coatings with active antibacterial activity have been obtained and investigated. Depending on where they will be used, coatings may need to have certain properties like antibacterial effectiveness, ability to work alongside biological systems, form, crystal structure, resistance to chemical changes, and more. Material selection must also account for synthesis temperature and growth rate. Table 3 compares different antibacterial ALD materials. The comparison of different coatings in terms of effectiveness shows that among oxides, zinc oxide [54,55,58] is the most effective. The antibacterial activity of ZnO is based on several independent mechanisms (see Section 3.3.1). Therefore, ZnO is the most versatile material and the most commonly used for ALD antibacterial coatings. Furthermore, ALD of ZnO can be performed at low temperatures starting from 20 °C [81], and its growth rate reaches 0.19–0.20 nm per cycle [36,135], which is the highest value for all ALD coatings studied. Disadvantages of ZnO include low biocompatibility and low stability in an aqueous medium, which limits its potential applications in water purification and disinfection. The effects of these drawbacks can be reduced or eliminated by doping or synthesizing complex ZnO-based oxide systems.

In contrast, TiO_2_ has excellent biocompatibility and stability, and it grows successfully at low temperatures but has a slower growth rate (0.03–0.06 nm per cycle, depending on the synthesis temperature and the type of precursor and substrate). However, the main disadvantage of titania is its much lower antibacterial activity compared to ZnO [54,55,58]. Titanium oxide is bactericidal only in the presence of ambient light (mainly UV), but it can also be bacteriostatic in the dark. Bacteriostatic TiO_2_ coatings are biocompatible and non-toxic to the organism and may therefore be preferable for implant applications. The problem of low antibacterial activity of TiO_2_ can be solved by doping, synthesis of mixed oxide systems, and noble metal sensitization (see Section 3.2).

Iron oxide (Fe_2_O_3_), like titanium oxide, is not an active bactericidal oxide, but in the form of coatings for MOFs it shows high photocatalytic activity and can be considered to be a promising antibacterial material. However, in our opinion, iron oxide is less promising than titanium oxide because of the difficulties in growing it via ALD. Other oxides such as ZrO_2_, Al_2_O_3_, and HfO_2_ are also very easily obtained via ALD, but they are expected to have weak antibacterial activity mainly due to their bacteriostatic effect and, in our opinion, have limited prospects of application in areas where specific properties such as bioinertness (Al_2_O_3_) or high mechanical properties (ZrO_2_) are required.

Metallic silver, like zinc oxide, is a very potent bactericidal material with multiple mechanisms of action. Unlike oxides, silver is grown via ALD in the form of nanoparticles, resulting in high Ag^+^ release. In particular, ALD Ag has been reported to have high antibacterial activity against *S. aureus* and MRSA, while being non-toxic for eukaryotic cells in vitro and in vivo. However, synthesis difficulties (unstable precursors, its low vapor pressure, low reactivity, and need to use non-standard co-reagents) limit the prospects of using this method to produce silver-based antibacterial coatings via ALD.

When comparing different antibacterial ALD coatings, it is very important to consider the influence of the substrate. The type and morphology of the substrate determines the morphology of the coatings, and for coatings with a thickness of a few tens of nanometers or less, wettability, surface charge, and surface energy are influential. However, the influence of the substrate is not limited to this. Because ALD is a process based on reactions between chemical groups on the substrate surface and gaseous reagents, the chemical composition of the substrate surface determines which functional groups will react. For example, there are many hydroxyl groups on the surface of cotton and silk, and only amide groups on polyamide [58], which can greatly affect the growth rate and continuity of the coating. For example, the samples of ALD TiO_2_-coated polyamide-66 substrates contain the nitrogen peaks in the X-ray photoelectron spectroscopy (XPS) spectra, indicating a non-cohesive film or the presence of a thinner film than expected. At the same time, for ZnO coatings, nitrogen is not observed in the XPS spectra, which means that the coating is continuous [58]. In the case of metals, even on inorganic substrates with a large number of hydroxyl groups on the surface, the density of growing NPs can vary greatly (Figure 15).

### 5.2. Advantages, Disadvantages, and Perspectives of ALD

The main advantages and disadvantages of the ALD method have been described in numerous review articles [18,20,22,137,138] and these remarks are applicable when considering the application of ALD for the synthesis of antibacterial coatings. All oxide films deposited are characterized by high uniformity and conformality. These properties make ALD very promising for coating surfaces with complex topography, such as 3D implants and scaffolds. In this context, ALD coatings are very promising for regulating the biocorrosion rate of biodegradable implants as shown by Li et al. [135]. Drug release control and active antibacterial ion release control can also be achieved by applying high quality ALD coatings [139]. Membranes and filters for water and air purification have high specific surface areas and also require the use of coating technologies with high uniformity and conformality.

Another feature of ALD is the cyclic nature of the growth and the small growth per cycle, which allows the thickness of the films to be controlled with high precision. This advantage is particularly important when considering zinc oxide and silver coatings, as the antibacterial effect and biocompatibility are highly dependent on the thickness and particle size of the coating. In addition, for a number of applications, such as nanozymes, it is necessary to control the particle size/thickness as precisely as possible. For example, in the study of Yang et al., the optimal size of platinum nanoparticles for Pt/CNTs composites was sought for a range of particle sizes in the range 0.55–2.81 nm, which is difficult to achieve by methods other than ALD [128].

Due to the low growth per cycle, ALD is not suitable for producing films with thicknesses greater than a few hundred nanometers, and other methods should be used to produce relatively thick films and structures. However, even in this case, ALD can be effective for the synthesis of seed layers. Indeed, in a large number of studies, ALD TiO_2_ [39] and ZnO [88,134,136] have been used as effective methods to create a homogeneous surfaces on which thicker layers of arrays of nanorods and other structures have been deposited (Figure 19), which due to their morphology and high specific surface area can be used as effective antibacterial ultrafiltration membranes, air filters, or implants.

It should also be noted that in addition to their application value, uniform and conformal ALD coatings can play an important role in basic research to identify and separate the various factors that determine different antibacterial effects. For example, Jeong et al. used 15 and 30 nm Al_2_O_3_ layers to regulate and separate the effect of chemical action of ZnO rods and their morphology on antibacterial properties [83].

When considering ALD coatings, it is rarely mentioned that the high adhesion of such coatings compared to coatings produced by other methods is a very important feature for many applications. The high adhesion is due to the chemical bond between the coating materials and the substrate. In addition, the excellent mechanical properties of ALD coatings have been noted. In particular, Darwish et al. studied ALD TiO_2_ coating on PMMA using the denture cleanser challenge test and showed the high stability, mechanical strength, and adhesion of titanium oxide. According to Konopatsky et al. [69], low adhesion is one of the main problems of noble metal-sensitized metal nanoparticle TiO_2_, as noble metals obtained via liquid phase deposition suffer from the gradual removal from the surface and reduce the efficiency of photocatalysis. In our recent work [57], we have successfully obtained similar hybrid structures of TiO_2_ nanocoatings/Ag NPs using only ALD. Such structures should exhibit higher adhesion. Thus, we have shown that such complex structures can be obtained in a single ALD process, which simplifies the synthesis and reduces the risk of poor adhesion of Ag nanoparticles [69] and is likely to increase interest in such structures in the future.

The ALD of mixed oxide systems, as well as multilayer or complex oxide–metal structures, has great research potential and may yield new promising results. In this direction, encouraging results have already been obtained in recent years, showing the synergistic effect of combinations of different oxides [63,64] and noble metal sensitization [30,60]. The ALD opens up great prospects for the synthesis of a wide variety of coatings of different compositions due to the possibility to flexibly vary the reagents used at different stages of the coating growth. Unfortunately, most studies so far use common bacteria like *E. coli* and *S. aureus*. For specific applications, more detailed studies are needed using specific strains. To develop coatings for medical implants, it is necessary to conduct more detailed studies in vivo. Finally, it is necessary to briefly discuss the prospects for the practical application of the ALD technologies and approaches described in this review. Most of the work presented requires additional reproducibility testing as well as additional in vitro and in vivo studies; some of the work requires the investigation of a number of other functional properties. Nevertheless, many of the coatings obtained can be used in practice and in industry after careful evaluation of their cost effectiveness. Although ALD has already become an industrial method in microelectronics and is even successfully used in the medical industry for surface modification of titanium implants [140,141], ALD is still a very expensive and complex technology with a very low coating growth rate. However, technical advances in recent years suggest that the impact of these negative factors can be reduced. In particular, we should note the development of technically simpler technologies such as atmospheric pressure ALD [142,143] and faster roll-to-roll ALD, spatial ALD [19], as well as the emergence of techniques that avoid the use of plasma to produce silver [123,144]. We believe that the advancement of these technologies could greatly augment the likelihood of extensive application of ALD in the manufacturing of antibacterial-coated materials within the industry.

## 6. Conclusions

This review details the results of using ALD for the synthesis of antibacterial coatings based on metals and oxides. The results of the analysis showed that the ALD method can be successfully applied to obtain a variety of different antibacterial nanocoatings on various inorganic and polymeric substrates. The influence of coating preparation conditions and their properties such as thickness, crystal structure, morphology, surface charge, wettability, etc., on the antibacterial properties was discussed in detail. It has been shown that for oxide coatings, increasing the thickness improves the antibacterial effect up to a certain limit, while for zinc oxide, increasing the thickness can induce a cytotoxic effect on eukaryotic cells. The surface topography, its charge, and wettability are closely related and also have a significant influence on the antibacterial properties, mainly by achieving a bacteriostatic effect. It was also found that ALD coatings with a thickness typical for this method (not more than a few tens of nanometers) are characterized by a significant influence of the substrate. In addition, a significant influence of the ambient illumination factor on the bactericidal activity of photocatalytic coatings based on pure titanium and zinc oxides, as well as mixed and complex systems based on them, was shown. The analysis of the results showed that mixed oxide systems and composite materials based on oxides and noble metals are the most promising as antibacterial ALD coatings for a wide range of applications.

The results of the review showed that ALD coatings have wide application prospects due to ALD features such as high uniformity, conformality, precise composition and thickness control, and high adhesion. In particular, the most active and successful research is in the areas of antibacterial ALD nanocoatings for dental and orthopedic implants, water and air purification systems, and antibacterial textiles. Some progress is also being made in the field of ultra-thin transparent coatings for touch screens and the treatment of various diseases and injuries.

## Figures and Tables

**Figure 1 antibiotics-12-01656-f001:**
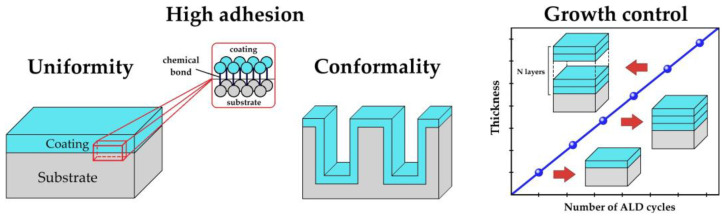
Diagram showing the main benefits of ALD.

**Figure 2 antibiotics-12-01656-f002:**
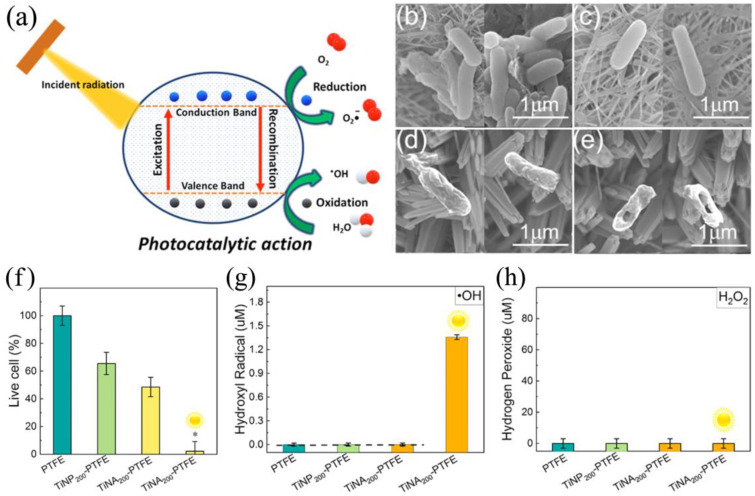
(**a**) The mechanism of the antibacterial activity of titanium oxide. Adapted with permission from Ref. [47]. (**b**–**e**) Antibacterial activity of the pristine polytetrafluoroethylene (PTFE) and TiO_2_ composite membranes. Representative SEM images of *E. coli* cells on (**b**) pristine PTFE membrane, TiO_2_ nanoparticle-PTFE composite membrane, 200 ALD cycles (TiNP_200_-PTFE membrane), (**d**) TiO_2_ nanoarray-PTFE composite membrane, 200 ALD cycles (TiNA_200_-PTFE membrane), and (**e**) TiNA200-PTFE membrane with sunlight irradiation. (**f**) Relative live cell numbers of *E. coli* cells after 3 h contact with different composite membranes, determined by plate-counting CFUs, normalized to the results of the PTFE membrane that had been presoaked in alcohol, and rinsed for full membrane wetting. Values marked with an asterisk (*) are significantly different from the value of the random sample (*n* = 3; *p* < 0.05). Under a solar simulator, the illumination intensity was ∼8 × 10^4^ lux. (**g**,**h**) Analysis of antibacterial mechanisms for pristine PTFE and TiO_2_ composite membranes: (**g**) cumulative hydroxyl radical generation over time indicated by the formation of fluorescent 2-hydroxyterephthalic acid (hTPA) from the reaction with terephthalate; (**h**) H_2_O_2_ concentration measured by red-fluorescing Amplex Red Assay (10-acetyl-3,7-dihydroxyphenoxazine in the presence of horseradish peroxidase). Adapted with permission from Ref. [39]. Copyright 2021 American Chemical Society.

**Figure 3 antibiotics-12-01656-f003:**
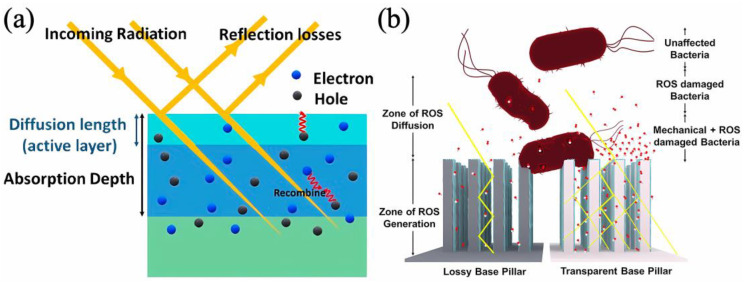
(**a**) The two loss mechanisms in planar photocatalytic films: reflection of incident light and recombination of photogenerated charge carriers. Only excitons generated within a few diffusion lengths from the surface (active layer) reach the surface. (**b**) TiO_2_-coated nanopillars that generate ROS that must diffuse out of the nanostructured valleys before it can damage bacteria. The optical properties of nanopillars material (base pillar) affect light absorption in photocatalytic coating. Adapted with permission from Ref. [50]. Copyright 2020 American Chemical Society.

**Figure 4 antibiotics-12-01656-f004:**
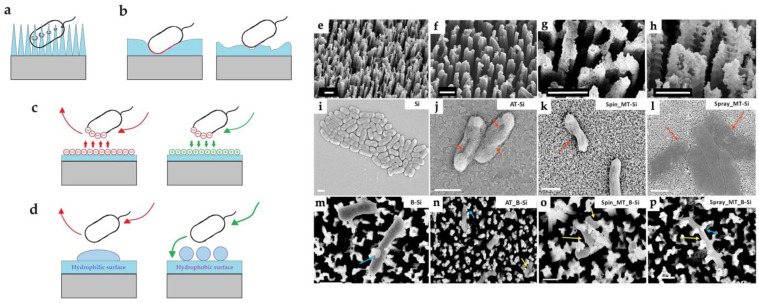
General strategies for the development of antibacterial surfaces are based on (**a**) contact killing, (**b**) the area of the contact zones, (**c**) electrostatic, and (**d**) wetting of the surface; (**e**–**h**) 45° view SEM images: (**e**) Black Si (B-Si) and (**f**) ALD-coated B-Si (AT_B-Si); mesoporous nanoparticles: (**g**) spin-coated (Spin_MT_B-Si) and (**h**) spray-coated (Spray_MT_B-Si) on B-Si (Scale bar: 1 μm). (**i**–**p**) Scanning electron microscopy images of bacteria on different substrates: (**i**) undamaged bacteria on flat Si; (**j**–**l**) bacterial cell wall disruption on ALD-, spin-, and spray-coated TiO_2_ surfaces, respectively; (**m**) bacterial cell has been pierced by nanostructures of black Si; (**n**–**p**) bacteria have sunken inside the TiO_2_-coated pillars, indicating cell death (Scale bar: 1 μm). Blue arrows indicate the piercing of bacteria by pillars. Red arrows indicate cell wall damaged areas. Yellow arrows indicate sunken bacteria. Adapted with permission from Ref. [47].

**Figure 5 antibiotics-12-01656-f005:**
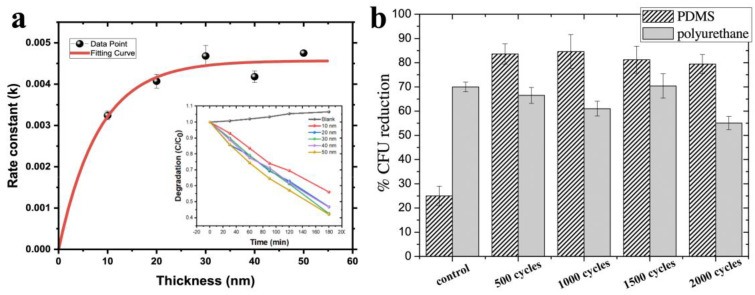
(**a**) Effect of film thickness on photocatalytic activity of TiO_2_. Reprinted with permission from Ref. [50]. Copyright 2020 American Chemical Society. (**b**) Percentage of reduction in the colonies of *C. albicans* on non-covered and TiO_2_-covered substrates after UV exposure. All the experiments were conducted in triplicate with average error within 5%. Adapted with permission from Ref. [29]. Copyright 2017 Elsevier.

**Figure 6 antibiotics-12-01656-f006:**
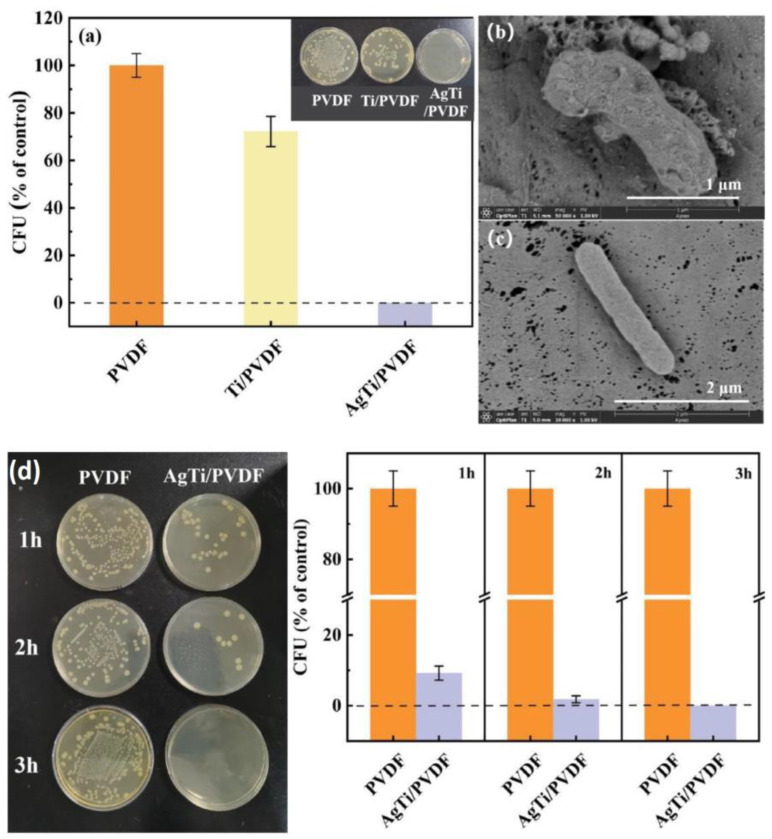
(**a**) The relative viability rate of bacteria after contact with the hybrid membrane surface for 3 h. (**b**) SEM image of bacterial morphology after contact with the hybrid membrane surface for 3 h. (**c**) SEM image of bacterial morphology on a PVDF membrane surface. (**d**) The images and the relative viable bacteria rate of *E. coli* exposed to PVDF and AgTi/PVDF hybrid membranes for 1 h, 2 h, and 3 h, respectively. Adapted with permission from Ref. [68]. Copyright 2022 Elsevier.

**Figure 7 antibiotics-12-01656-f007:**
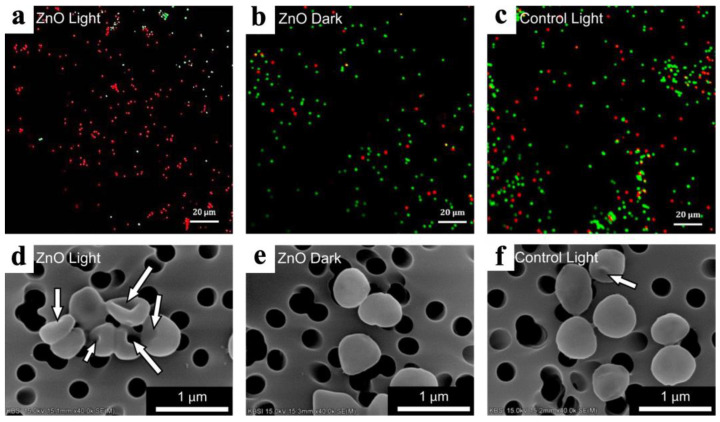
Microscopic images of *S. aureus* cells under the three conditions: the glass substrate without ZnO film was exposed to UV light (Control Light), the ALD ZnO specimen was not exposed to UV light (ZnO Dark), and the ALD ZnO specimen was exposed to UV light (ZnO Light). (**a**–**c**) are CLSM images. Cells with damaged membranes appear in red hue, while intact cells appear in green hue. (**d**–**f**) are field emission scanning electron microscopy (FESEM) images. Arrows indicate cells with damaged membranes. Reprinted with permission from Ref. [70]. Copyright 2017 Elsevier.

**Figure 8 antibiotics-12-01656-f008:**
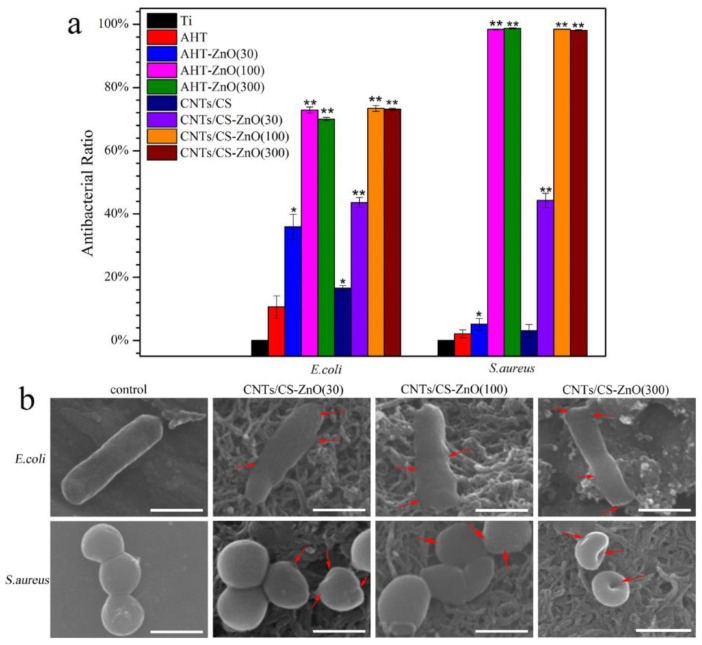
(**a**,**b**) Antibacterial activity of different samples against *E. coli* and *S. aureus*. (**a**) Antibacterial ratio of Ti, alkali-heat-treated Ti (AHT), carbon nanotubes/chitosan composites (CNTs/CS), CNTs/CS-ZnO(30), CNTs/CS-ZnO(100), and CNTs/CS- ZnO(300) against *E. coli* and *S. aureus*. (**b**) SEM images of the attachment of *E. coli* and *S. aureus* cells to the untreated Ti surface, CNTs/CS-ZnO(30), CNTs/CS-ZnO(100), and CNTs/CS-ZnO(300) after incubation at 37 °C for 12 and 24 h, respectively. Bacteria exhibiting ribbed structures and compromised membranes are highlighted with red arrows. The scale bar is 1 μm. * *p* < 0.05 and ** *p* < 0.01 versus the Ti group. Reprinted with permission from Ref. [43]. Copyright 2016 Elsevier.

**Figure 9 antibiotics-12-01656-f009:**
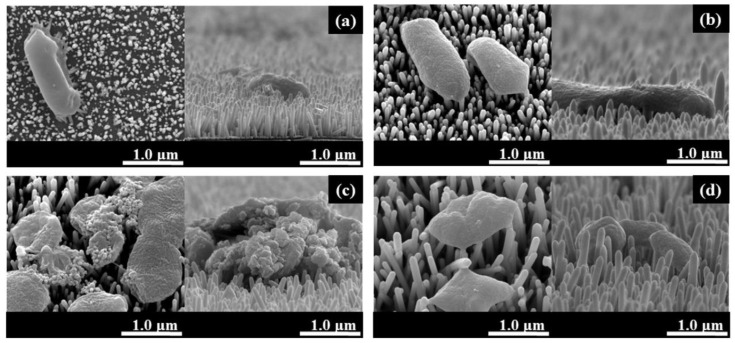
Top view FE-SEM images of a ZnO NR array surface after inactivation of *E. coli* under dark conditions. The length of the arrays was (**a**) 0.5 μm, (**b**) 1 μm, (**c**) 2 μm, and (**d**) 4 μm. Reprinted with permission from Ref. [83]. Copyright 2020 Elsevier.

**Figure 10 antibiotics-12-01656-f010:**
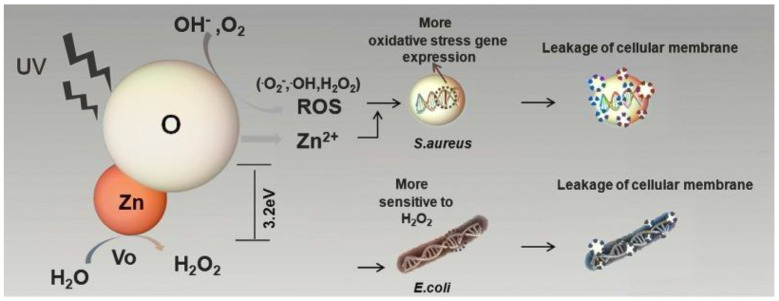
Schematic illustration of the differences in the antibacterial mechanisms between two atomic layers of ZnO (ZnO-O top, ZnO-Zn bottom). Adapted with permission from Ref. [79]. Copyright 2016 Wiley-VCH Verlag GmbH & Co. KGaA, Weinheim.

**Figure 11 antibiotics-12-01656-f011:**
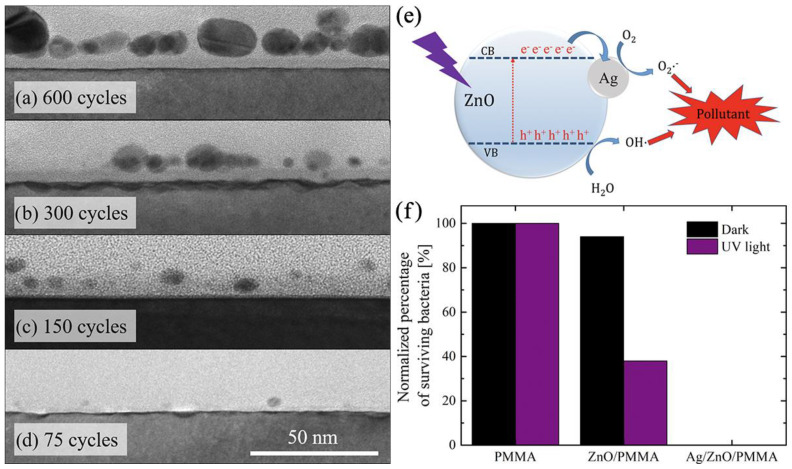
(**a**–**d**) TEM images, in cross view, of Ag nanoparticles deposited on ZnO coatings and Si substrates using 80 °C and varying cycles: (**a**) 600, (**b**) 300, (**c**) 150, and (**d**) 75. (**e**) Electron capture by Ag nanoparticles in contact with ZnO. (**f**) Normalized percentage of surviving *E. coli* bacteria in the dark and when exposed to UV irradiation in contact with PMMA, ZnO/PMMA, and Ag/ZnO/PMMA samples. The experimental error was around 5%. Experiments were repeated three times. Adapted with permission from Ref. [38].

**Figure 12 antibiotics-12-01656-f012:**
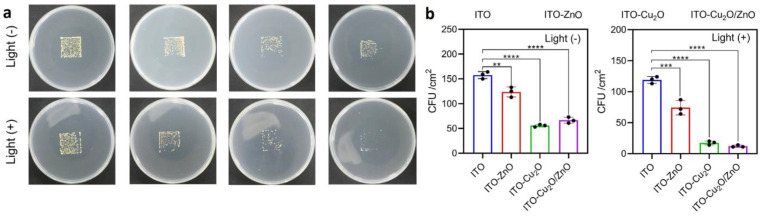
(**a**) Spread plates and (**b**) corresponding strain counts of ITO, ITO-Cu_2_O, ITO-ZnO, and ITO-Cu_2_O/ZnO in the absence and presence of solar light. The error bar means ± standard deviations (*n* = 3 independent samples): ** *p* < 0.01, *** *p* < 0.001, **** *p* < 0.0001. Adapted with permission from Ref. [31]. Copyright 2022 Published by Elsevier Ltd.

**Figure 13 antibiotics-12-01656-f013:**
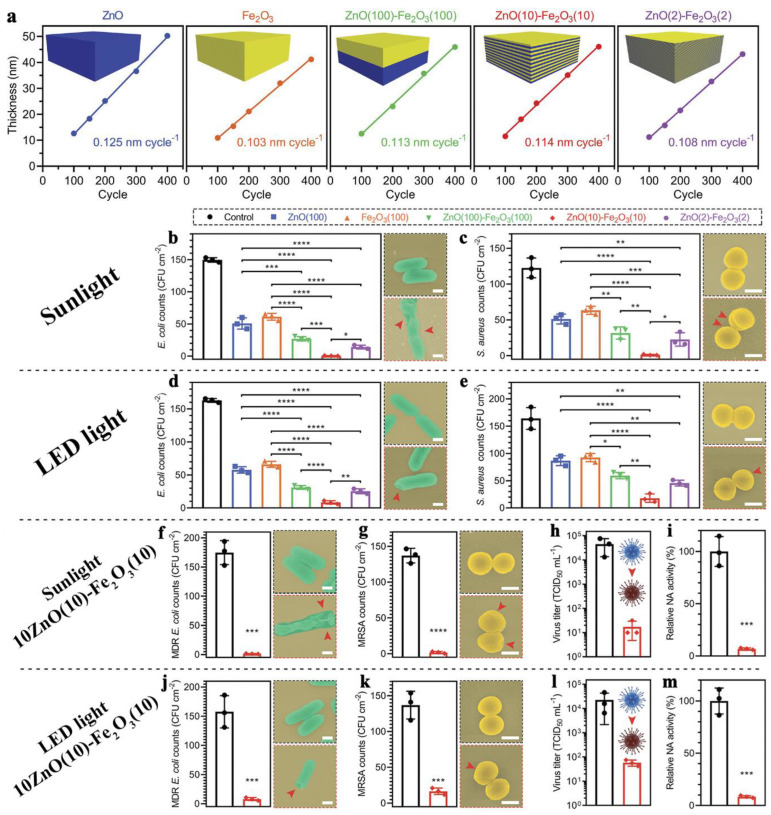
(**a**) Thickness measurements of ZnO, Fe_2_O_3_, ZnO(100)–Fe_2_O_3_(100), ZnO(10)–Fe_2_O_3_(10), and ZnO(2)–Fe_2_O_3_(2) with different numbers of cycles via spectroscopic ellipsometry. (**b**–**m**) Photocatalytic disinfection performance. (**b**) *E. coli* counts (all bacterial counts were achieved by using the bacterial imprinting method) in control, ZnO(100), Fe_2_O_3_(100), ZnO(100)–Fe_2_O_3_(100), ZnO(10)–Fe_2_O_3_(10), and ZnO(2)–Fe_2_O_3_(2) groups under sunlight irradiation (1 sun) for 2 min and corresponding SEM morphologies (scale bars = 500 nm for all SEM images) of *E. coli* in control and ZnO(10)–Fe_2_O_3_(10) groups (they were marked with black color and red color, respectively). (**c**) *S. aureus* counts in control, ZnO(100), Fe_2_O_3_(100), ZnO(100)–Fe_2_O_3_(100), ZnO(10)–Fe_2_O_3_(10), and ZnO(2)–Fe_2_O_3_(2) groups under sunlight irradiation (1 sun) for 4 min and corresponding SEM morphologies of *S. aureus* in control and ZnO(10)–Fe_2_O_3_(10) groups. (**d**) *E. coli* counts in control, ZnO(100), Fe_2_O_3_(100), ZnO(100)–Fe_2_O_3_(100), ZnO(10)–Fe_2_O_3_(10), and ZnO(2)–Fe_2_O_3_(2) groups under LED light irradiation (2 W m^−2^) for 30 min and corresponding SEM morphologies of *E. coli* in control and ZnO(10)–Fe_2_O_3_(10) groups. (**e**) *S. aureus* counts in control, ZnO(100), Fe_2_O_3_(100), ZnO(100)–Fe_2_O_3_(100), ZnO(10)–Fe_2_O_3_(10), and ZnO(2)–Fe_2_O_3_(2) groups under LED light irradiation (2 W m^−2^) for 60 min and corresponding SEM morphologies of *S. aureus* in control and ZnO(10)–Fe_2_O_3_(10) groups. (**f**) MDR *E. coli* counts and corresponding SEM morphologies in control and ZnO(10)–Fe_2_O_3_(10) groups under sunlight irradiation (1 sun) for 2 min. (**g**) MRSA counts and corresponding SEM morphologies in control and ZnO(10)–Fe_2_O_3_(10) groups under sunlight irradiation (1 sun) for 4 min. (**h**) H1N1 virus titer and (**i**) relative NA activity in control and ZnO(10)–Fe_2_O_3_(10) groups under sunlight irradiation (1 sun) for 4 min. (**j**) MDR *E. coli* counts and corresponding SEM morphologies in control and ZnO(10)–Fe_2_O_3_(10) groups under LED light irradiation (2 W m^−2^) for 30 min. (**k**) MRSA counts and corresponding SEM morphologies (scale bars = 500 nm for all SEM images) in control and ZnO(10)–Fe_2_O_3_(10) groups under LED light irradiation (2 W m^−2^) for 60 min. (**l**) H1N1 virus titer and (**m**) relative NA activity in control and ZnO(10)–Fe_2_O_3_(10) groups under LED light irradiation (2 W m^−2^) for 60 min. The red arrows in SEM images mark obvious damage and deformation of bacteria. Individual data points (*n* = 3 biologically independent samples) and the error bars indicate means ± standard deviations. Statistical differences were analyzed through one-way ANOVA and post hoc Tukey’s test or *t*-test (* *p* < 0.05, ** *p* < 0.01, *** *p* < 0.001, **** *p* < 0.0001). Adapted with permission from Ref. [92]. Copyright 2023 Wiley-VCH GmbH.

**Figure 14 antibiotics-12-01656-f014:**
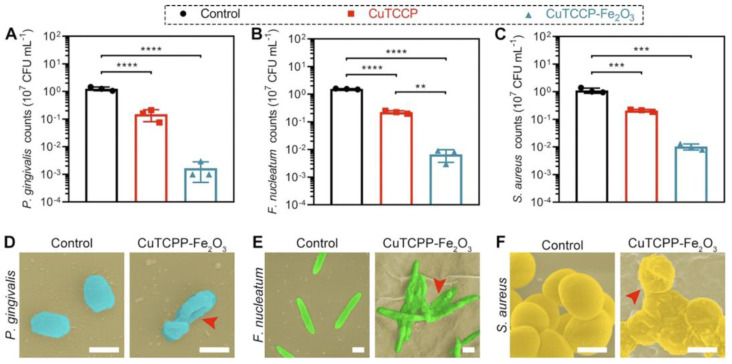
Antibacterial mechanism of photodynamic ion therapy. Viabilities of (**A**) P. *gingivalis*, (**B**) F. *nucleatum*, and (**C**) *S. aureus* treated with CuTCPP and CuTCPP-Fe_2_O_3_ under 660 nm laser illumination for 20 min, followed by incubation in the dark for 2 h. SEM images (scale bars = 500 nm for all images) and corresponding EDS curves of (**D**) P. *gingivalis*, (**E**) F. *nucleatum*, and (**F**) *S. aureus* treated with CuTCPP-Fe_2_O_3_ under 660 nm light illumination for 20 min, followed by incubation in the dark for 2 h. Red arrows show obvious damage and deformation of bacteria. Individual data points (*n* = 3 biologically independent samples) and error bars indicate means ± standard deviations. Statistical differences were analyzed through one-way ANOVA and post hoc Tukey’s test (** *p* < 0.01, *** *p* < 0.001, **** *p* < 0.0001). Adapted with permission from Ref. [97]. Copyright 2021 American Chemical Society.

**Figure 15 antibiotics-12-01656-f015:**
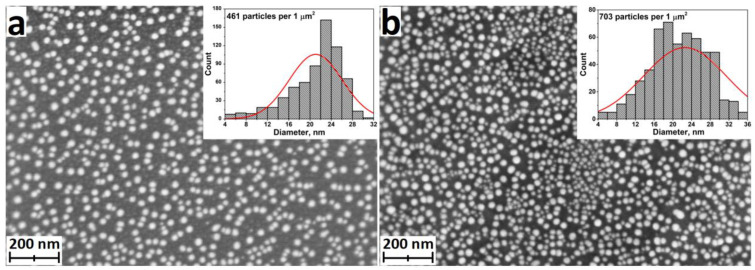
SEM images of Ag nanoparticles deposited on the silicon surface (**a**) and titanium disks (**b**). Reprinted with permission from Ref. [57].

**Figure 16 antibiotics-12-01656-f016:**
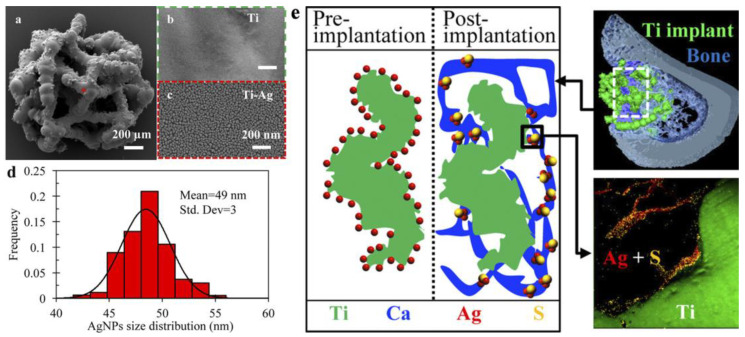
(**a**) Macroscopic SEM image of the additively manufactured porous titanium scaffold. High-resolution SEM images of (**b**) titanium scaffolds and (**c**) silver-coated titanium scaffolds. (**d**) Size distribution histogram of silver particles (125 °C for 500 ALD cycles). (**e**) The nanoscale elemental mapping of the bone-implant interface showed that silver was present primarily in the osseous tissue and colocalized with sulfur. TEM revealed silver sulfide nanoparticles in the newly regenerated bone, presenting strong evidence that the previously in vitro observed biotransformation of silver to silver sulfide occurs in vivo. Adapted with permission from Ref. [27].

**Figure 17 antibiotics-12-01656-f017:**
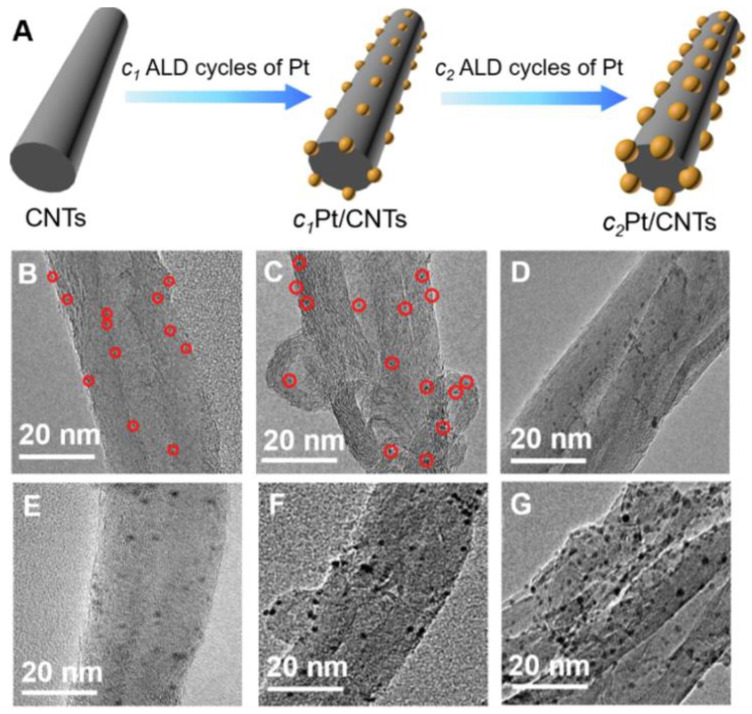
Synthesis and structural characterizations of Pt/CNTs nanozymes. (**A**) Illustration of the preparation process of Pt/CNTs nanozymes. (**B**–**G**) TEM images of 5Pt/CNTs, 10Pt/CNTs, 20Pt/CNTs, 30Pt/CNTs, 50Pt/CNTs, and 70Pt/CNTs, respectively. Note: red circles refer to Pt nanoparticles. Adapted with permission from Ref. [128].

**Figure 18 antibiotics-12-01656-f018:**
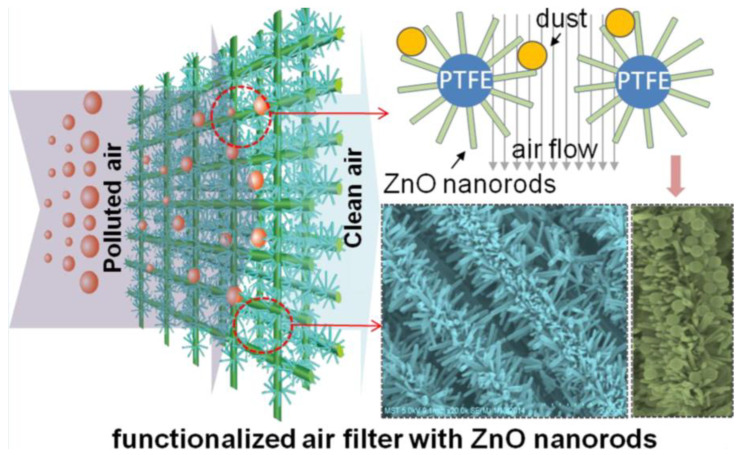
Schematic of the functionalized air filter with ZnO nanorods (**left**). Schematic of the dust particle capture mechanism and SEM images of the functionalized air filter with ZnO nanorods (**right**). Reprinted with permission from Ref. [88]. Copyright 2015 American Chemical Society.

**Figure 19 antibiotics-12-01656-f019:**
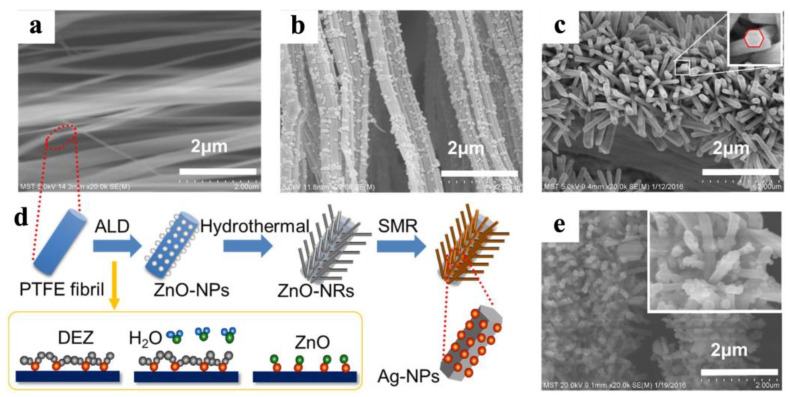
Schematic of the preparation process of the Ag@ZnO-functionalized PTFE filters and the SEM images of the filters at different stages of production: (**a**) pristine PTFE, (**b**) ZnONPs@ PTFE, (**c**) ZnO-NRs@PTFE, (**d**) schematic of the preparation of Ag/ZnO-NRs@PTFE, and (**e**) Ag/ZnO-NRs@PTFE; the inset is a high magnification micrograph of Ag/ZnO-NRs@PTFE. Reprinted with permission from Ref. [91]. Copyright 2017 Elsevier.

**Table 1 antibiotics-12-01656-t001:** Results of the study of antibacterial ALD TiO_2_ coatings.

Motivation	Support/Precursors	ALD Temperature/Thickness	Bacteria/Irradiation	Results	Ref.
Examine antibacterial coatings for purifying water.	Si/TiCl_4_/H_2_O	100, 200, 300 °C/50 nm	*E. coli*/ UV irradiation	Increasing ALD temperature from 200 °C to 300 °C improves TiO_2_ photocatalytic activity but does not have an impact on antibacterial activity. *E. coli* survival: 200 °C–61%, 300 °C–66%.	[48]
Study the effect of coatings with different morphology and structure on osteoblast, fibroblast functions, and bacterial activities.	Ti/TDMAT/H_2_O	120, 160, 190 °C/100 nm	*S. aureus*, *E. coli*, MRSA	TiO_2_ inhibit adhesion and growth bacteria and fibroblast but improve osteoblasts adhesion and proliferation. The 160 °C sample (amorphous) showed the best antibacterial activity. An increase in nanoscale roughness and greater hydrophilicity contribute to increased protein adsorption, which may affect the cellular/bacterial activities.	[49]
The study of photocatalytic processes in antibacterial activity TiO_2_.	Black Si and SiO_2_ nanopillars/(no data)	No data/50 nm	*E. coli*	The increase in the light absorption does not lead to an increase in ROS production. TiO_2_-coated nanopillars arrays made of SiO_2_ have 73% higher bactericidal efficacies than those made of Si.	[50]
Study of the synergic effect present in mixed anatase/rutile TiO_2_ on antibacterial properties.	Rutile-TiO_2_ nanotubes (RTNT)/ TDMAT/H_2_O	250 °C Annealing—450 °C 2 h/10 nm	*E. coli/* UV light irradiation	A considerable increase in photocatalytic activity and ROS generation using RTNT coated with anatase-TiO_2_ compared to only rutile or anatase. A considerable increase in bactericidal activity using RTNT coated with A-TiO_2_ compared to single R or A-TiO_2_ nanotubes.	[45]
Examine surface coating of nanoporous Al_2_O_3_ for controlled pore size reduction.	Porous Al_2_O_3_ membranes/TiCl_4_/H_2_O	300 °C/4.3 and 8.6 nm	*S. aureus*, *E. coli*, UV irradiation	The 20 nm pore size TiO_2_-coated nanoporous alumina membrane inhibited microbial adhesion while the 100 nm pore size TiO_2_-coated membrane did not.	[51]
Examine the efficiency of a bacteria-resistant coating for the PDMS cochlear implants.	PDMS ^1^/TDMAT/O_2_ plasma	100 °C/(10–40 nm)	*E. coli*	TiO_2_-coated surfaces save the integrity of polymeric materials and reduce *E. coli* colonization and biofilm formation with protein quantity on ALD- and lymphoproliferative diseases (LPD) treated samples being reduced by 44 and 41%. Confocal laser scanning microscopy—biofilm reduction of 91% for ALD-coated surfaces.	[52]
Studies of antibacterial properties of TiO_2_ deposited on polymers used for catheters and contact lenses.	PU ^2^ and PDMS/ TiCl_4_/H_2_O	80 °C/50–200 nm	*Candida albicans* UV and without UV	TiO_2_ suppressed the yeast hyphal transition of *C. albicans* onto PU; however, a high adhesion of *C. albicans* was observed. For PDMS + TiO_2_, the yeast adhesion did not change, as observed in the control. After UV treatment, TiO_2_+PDMS had better reduction in the colony-forming unit CFU (up to 59.5%) compare to the uncoated PDMS, while no difference was observed in TiO_2_-covered PU.	[29]
Improve the surface characteristics of denture PMMA.	PMMA ^3^/TDMAT/O_3_	65 °C/30 nm	*Candida* *albicans*	*C. albicans* reduction reached 63% to 77% for the attachment test and 56% for biofilm formation.	[53]

^1^ PDMS—polydimethylsiloxane. ^2^ PU—polyurethane. ^3^ PMMA—polymethyl methacrylate.

**Table 2 antibiotics-12-01656-t002:** Characteristics of Ag ALD antibacterial coatings.

Support/ALD Conditions	Characteristics	Bacteria	Results	Ref.
TiO_2_ nanotubes 10–84 nm/25 cycles of plasma enhanced atomic layer deposition (PEALD) Ag(fod)(PEt_3_) ^1^ +H_2_ (120 °C).	7.8–9.2 nm dispersed Ag particles and dense Ag layer	*S. aureus*	Low level of Ag^+^ releasing the sample with the smallest nanotube diameter and a continuous layer of Ag showed the best bactericidal results.	[102]
3d Ti scaffolds/direct liquid injection of 0.1 M (hfac)Ag(1,5-COD) ^2^ in toluene and propan-1-ol. 500 ALD cycles (125 °C).	Effective layer—13 nm (NPs 40–90 nm diameter)	*S. aureus* (MRSA) *S. epidermidis.*	On bare titanium scaffolds, *S. epidermidis* growth was slow but on Ag-coated there were significant further reductions (two orders of inhibition) in both bacterial recovery and biofilm formation. MRSA growth was similarly slow on bare titanium and not further affected by Ag coating.	[122]
3d Ti scaffolds/direct liquid injection 0.1M (hfac)Ag(1,5-COD) in toluene and propan-1-ol. 500 ALD cycles (125 °C).	Average NPs 49 nm in diameter	In vivo—implants in rat tibial defects	No effect of Ag on bone formation and osseointegration after 2–12 weeks of implantation. Ag is a part of less toxic Ag_2_S within the newly formed bone tissue adjacent to the implant surface.	[27]
Ti/ALD TiO_2_ (TiCl_4_ + H_2_O) + Ag (Ag(fod)(PEt_3_) + H_2_).	20–28 nm on Si 16–30 nm on Ti	*S. aureus*	The ALD combination of TiO_2_ + Ag is significantly more active against *S. aureus* than pure TiO_2_ and Ag.	[57]
N95 medical mask (PP, polyester)/Ag(fod)(PEt_3_) + (CH_3_)_2_NH*BH_3._	16 nm	*S. aureus*	A 76% reduction in *S. aureus* CFU was observed after 24 h. At an early stage (2 h), Ag had no bactericidal effect.	[123]

^1^ (2,2-dimethyl-6,6,7,7,8,8,8-heptafluorooctane-3,5-dionato) silver(I) triethyl-phosphine. ^2^ (hexafluoroacetylacetonato) silver(I)(1,5-cyclooctadiene).

**Table 3 antibiotics-12-01656-t003:** Growth and functional characteristics of antibacterial ALD coatings.

Characteristics	TiO_2_	Doped TiO_2_	ZnO	Ag	ZrO_2_	Fe_2_O_3_	Al_2_O_3_
Antibacterial efficacy	+	++	+++	+++	0	+	0
Biocompatibility	+++	++	+	+	++	+	+
ALD temperature	From 65 °C	From 65 °C	From RT	120–160 °C	~200 °C	~180 °C	From RT
Growth rate, uniformity, and conformality	++	++	+++	+	++	++	+++
Morphology of the coating	Dense coating	Dense coating	Dense nanocrystalline coating	NPs	Dense coating	Dense coating	Dense coating
Stability in aqueous environment	+++	++	+	+	+++	+	+++

+++—maximum performance. ++—high performance. +—average performance. 0—minimal or no effectiveness. RT—room temperature.

## Data Availability

Not applicable.

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
