# Peer review of "Atomic Layer Deposition of Antibacterial Nanocoatings: A Review"

_antibiotics, 2023, doi:10.3390/antibiotics12121656_

Round 1

Reviewer 1 Report

Comments and Suggestions for Authors

This review on the ALD of antibacterial nanocoatings is well-written, well-organized and comprehensive enough.

It would be strengthened significantly if:

-- all abbreviations (e.g., PTFE, TINA, TINP, FESEM, AHT, …) are explained the first time they are used;

-- experimental uncertainties are shown in all data (e.g., see figure 12f);

-- its English is double-checked (e.g., nanocotings in pages 21 and 24);

-- spatial ALD is more prominently highlighted throughout the manuscript, since it seems to remove the perhaps main disadvantage of ALD: its slow rate. Spatial ALD is now barely mentioned once in the next to the last line, before the Conclusions section.

Comments on the Quality of English Language

The quality of English is good.  It needs some improvements as it is indicated in the comments to the authors section.

Author Response

Comments and Suggestions for Authors

This review on the ALD of antibacterial nanocoatings is well-written, well-organized and comprehensive enough.

It would be strengthened significantly if:

-- all abbreviations (e.g., PTFE, TINA, TINP, FESEM, AHT, …) are explained the first time they are used;

Thank you for your comments. All abbreviations are now explained in the text the first time they are used. The section on abbreviations has also been added at the end of the manuscript.

-- experimental uncertainties are shown in all data (e.g., see figure 12f);

Experimental uncertainties have now been added to all figure captions (Figures 5, 11, 13, 14 in the updated version of the manuscript) where these data were available.

-- its English is double-checked (e.g., nanocotings in pages 21 and 24);

The text has been proofread and some other errors corrected.

-- spatial ALD is more prominently highlighted throughout the manuscript, since it seems to remove the perhaps main disadvantage of ALD: its slow rate. Spatial ALD is now barely mentioned once in the next to the last line, before the Conclusions section.

Yes, spatial ALD is indeed a technology for the rapid deposition of nanoscale thin films. Due to the spatial separation of precursor gases using specially designed reactors, the slow purge steps and chamber requirements of conventional ALD are eliminated, allowing the deposition of conformal thin films over a large area and at high throughput.

Spatial ALD has been widely used to produce ZnO and a number of other oxides, but no studies have yet been carried out on antibacterial properties! Therefore, we mention this type of ALD only at the very end of the paper as a promising approach for scale-up and industrial production.

Comments on the Quality of English Language

The quality of English is good. It needs some improvements as it is indicated in the comments to the authors section.

A small English language revision has been made. Thank you very much!

Reviewer 2 Report

Comments and Suggestions for Authors

I have some suggestions as following for authors' reference,

first, for the searching method and resource, I suggest that authors should also consider the searching engine as explored by MDPI, "Scilit", www.scilit.net. 

second, I suggest that authors put the Method section in supplementary information, not shown up in the manuscript. 

third, I believe that the Results section was the key part in this review. The refence number beginning here was [47]. So, I wonder whether all references before reference [47] were necessary or not to be cited in the background. I suggest that authors should refine those references, since the total numbers of references here were only 144. 

forth, the section 3.1 and 3.2 might be merged into one, since they were all about TiO2. The similar situation happens also in section 3.3 and "the first" 3.4 (be careful, they have two "3.4" here). Also, why the second "3.4 other compounds" goes before section 3.5? It sounds like other compounds should be mentioned in the last section in this categorization system. So, I strongly suggest that authors should reorganize those categories. 

fifth, critical descriptions should be added in current section 3. I would like to know why some guys like to use TiO2, and why other guys used different basic materials. Why people kept pursuing new basic materials? What kind of advantages and disadvantages generated by using those diverse stuffs? It is highly desired in this kind review, instead of only presenting the contents from different literature. I also noticed that authors have "the first" section 5 (authors have two section 5 here) to handle this issue. But I suggest that it came too late. Authors should put valuable comments along with the introduction and description of detailed infomation about ALD materials. So, authors might have to put the words in "the first" section 5 into the exact section together with the exact examples in section 3 and 4. 

sixth, I would like to see the perspective section in the "second" section 5. The current conclusion summarizing the review is not enough. 

Comments on the Quality of English Language

I have some suggestions as following for authors' reference,

first, for the searching method and resource, I suggest that authors should also consider the searching engine as explored by MDPI, "Scilit", www.scilit.net. 

second, I suggest that authors put the Method section in supplementary information, not shown up in the manuscript. 

third, I believe that the Results section was the key part in this review. The refence number beginning here was [47]. So, I wonder whether all references before reference [47] were necessary or not to be cited in the background. I suggest that authors should refine those references, since the total numbers of references here were only 144. 

forth, the section 3.1 and 3.2 might be merged into one, since they were all about TiO2. The similar situation happens also in section 3.3 and "the first" 3.4 (be careful, they have two "3.4" here). Also, why the second "3.4 other compounds" goes before section 3.5? It sounds like other compounds should be mentioned in the last section in this categorization system. So, I strongly suggest that authors should reorganize those categories. 

fifth, critical descriptions should be added in current section 3. I would like to know why some guys like to use TiO2, and why other guys used different basic materials. Why people kept pursuing new basic materials? What kind of advantages and disadvantages generated by using those diverse stuffs? It is highly desired in this kind review, instead of only presenting the contents from different literature. I also noticed that authors have "the first" section 5 (authors have two section 5 here) to handle this issue. But I suggest that it came too late. Authors should put valuable comments along with the introduction and description of detailed infomation about ALD materials. So, authors might have to put the words in "the first" section 5 into the exact section together with the exact examples in section 3 and 4. 

sixth, I would like to see the perspective section in the "second" section 5. The current conclusion summarizing the review is not enough. 

Author Response

I have some suggestions as following for authors' reference,

first, for the searching method and resource, I suggest that authors should also consider the searching engine as explored by MDPI, "Scilit", www.scilit.net.

Thank you for the link. We did a search using Scilit and got similar results to our first search of the Scopus database. Scilit gives 161 articles. Scopus gives 148. Most of these are not relevant. We didn't find any new relevant studies using Scilit.

second, I suggest that authors put the Method section in supplementary information, not shown up in the manuscript.

Yes, we fully agree with you. We have moved most of the Methods section to Supplementary Information.

third, I believe that the Results section was the key part in this review. The refence number beginning here was [47]. So, I wonder whether all references before reference [47] were necessary or not to be cited in the background. I suggest that authors should refine those references, since the total numbers of references here were only 144.

Thank you for this comment. Undoubtedly, the most important part of this review is the results section, as it contains the most important information about the currently available research in the field of synthesis of antibacterial coatings by ALD.

In fact, only the first 25 references are not directly related to the results section. The point is that we have already mentioned some of the articles that make up the results section in the introduction. At the same time, the first 25 references are mainly contemporary reviews related to the topic of our review (not 47). These articles allow the reader, who is not very familiar with the subject of the review, to find the necessary information and to understand it.

Regarding the total number of references, only 68 relevant articles were published at the time of the review, as described in the Methods section (now in the Supplementary Information). It turns out that about half of the references in our review are directly related to the topic of the article (this is the main core of the review), and the other half are related references that allow the reader to become more familiar with the questions that arise when reading the review, or they help to explain the results obtained in the main (core) articles. In our opinion, this ratio is optimal for an in-depth review article.

forth, the section 3.1 and 3.2 might be merged into one, since they were all about TiO2. The similar situation happens also in section 3.3 and "the first" 3.4 (be careful, they have two "3.4" here). Also, why the second "3.4 other compounds" goes before section 3.5? It sounds like other compounds should be mentioned in the last section in this categorization system. So, I strongly suggest that authors should reorganize those categories.

Pure (pristine) titanium oxide and doped titanium oxide are completely different substances. They also have very different characteristic of the surface antibacterial activity and its mechanisms. This is why we have separated these materials into a separate section. The same applies to zinc oxide. At the same time, mixed oxides can form complex structural systems and even nanolaminates, so we have included them in a separate section.

As the size of the review is relatively large, we have tried to create more sections to make it easier for the reader to navigate and not have to read a large amount of material if they only need to know narrow specific data.

As for the error in the numbering order, we have corrected it. Thank you very much!

For sections 3.4 and 3.5, it is appropriate to describe the outcomes for other usual ALD oxides and binary compounds after discussing pure and doped oxides. And then we should move on to metals, which are not strictly speaking compounds and grow in the form of particles rather than continuous layers. Therefore, we have placed silver and other metals in a separate section after discussing typical ALD compounds (presumably oxides).

To make this clearer, we have renamed the “Other Compounds” section to “Other Binary and More Complex Oxide Compounds”. We hope this is more accurate.

fifth, critical descriptions should be added in current section 3. I would like to know why some guys like to use TiO2, and why other guys used different basic materials. Why people kept pursuing new basic materials? What kind of advantages and disadvantages generated by using those diverse stuffs? It is highly desired in this kind review, instead of only presenting the contents from different literature. I also noticed that authors have "the first" section 5 (authors have two section 5 here) to handle this issue. But I suggest that it came too late. Authors should put valuable comments along with the introduction and description of detailed infomation about ALD materials. So, authors might have to put the words in "the first" section 5 into the exact section together with the exact examples in section 3 and 4.

Yes, we understand you very well. If you read the manuscript from the beginning, many questions arise. And many of these questions are only answered in the last sections.

But we structured the article based on the standard principles of research articles, including an introduction, methodology description, results description, and a section for discussion and analysis of the results. The PRISMA methodology, recommended by MDPI for systematic reviews, guided our approach. According our approach the results section aims to summarise only the available results, organised by the chemical composition of the materials and the effect of various coating properties such as thickness, structure and morphology. The subsequent section will examine how the coatings may be utilised whereas a general analysis of the results, comparing and providing prospects, will be presented only in the final section 5. We believe that this structure is both coherent and logical.

We believe that moving and duplicating summarising information from section 5 to sections 3 and 4 would unnecessarily increase the size of the review. But to help the reader navigate, we have included a description of the structure of the article in the abstract and at the end of the introduction.

Abstract - The aim of this paper is to carefully evaluate and analyze the relevant literature on this topic. Simple oxide coatings, including TiO2, ZnO, Fe2O3, MgO, and ZrO2, were examined, as well as coatings containing metal nanoparticles such as Ag, Cu, Pt, and Au, and mixed systems such as TiO2-ZnO, TiO2-ZrO2, ZnO-Al2O3, TiO2-Ag, and ZnO-Ag. Through comparative analysis, we have been able to draw conclusions on the effectiveness of various antibacterial coatings of different compositions, including key characteristics such as thickness, morphology, and crystal structure. The use of ALD in the development of antibacterial coatings for various applications was analyzed. Furthermore, assumptions were made about the most promising areas of development. The final section provides a comparison of different coatings, as well as the advantages, disadvantages and prospects of using ALD for the industrial production of antibacterial coatings.

Introduction - In this review, we provide a detailed analysis of the use of various inorganic coat-ings produced by ALD as antibacterial coatings. The results section of the review is structured according to the chemical composition of the coatings, but the following sections also provide a discussion of the results, with particular emphasis on areas of practical application, comparing the results for different compounds and analysing the influence of the above factors on the antibacterial performance. Finally, in the last section, an attempt is made to evaluate the future prospects of ALD.

And we hope it will make it easier for the reader to find the information they need.

As for your specific questions, the answers are as follows:

1) Material selection - Depending on the application, coatings with a specific set of functional characteristics such as level of antibacterial efficacy, biocompatibility, morphology, crystal structure, chemical stability, etc. are required. Material selection also requires consideration of synthesis temperature and growth rate. The comparative characterisation of various antibacterial ALD materials is given in Table 3 and different antibacterial compounds are compared in section 5.1 in detailed. We have edited the first paragraph of this section to make this clearer to the reader

2) Why search for new materials – The answer may seem obvious, but the existing materials are not optimal and therefore there is an active search both from the point of view of development of new materials and optimisation of methods of obtaining already known materials. We have slightly rewritten the first paragraph of the introduction and added to it to make this motivation clearer

3) Advantages and disadvantages - This is also reflected in Table 3 and its accompanying text.

sixth, I would like to see the perspective section in the "second" section 5. The current conclusion summarizing the review is not enough.

The discussion of perspectives and potential areas of investigation for antibacterial ALD coatings is the focus of the last two sections of Section 5.2. We assess the potential for further studies from both scientific and practical perspectives regarding ALD coatings for industrial applications. To the existing text we can only add a discussion on the need for more studies with more specific strains and in vivo studies in case of possible applications of coatings for medical implants. If necessary, please suggest additional topics for discussion of perspectives.

The revised outlook text is now:

The ALD of mixed oxide systems, as well as multilayer or complex oxide-metal structures, has great research potential and may yield new promising results. In this direction, encouraging results have already been obtained in recent years, showing the synergistic effect of combinations of different oxides [63,64] and noble metal sensitization [30,60]. The ALD opens up great prospects for the synthesis of a wide variety of coatings of different composition due to the possibility to flexibly vary the reagents used at different stages of the coating growth. Unfortunately, most studies so far use common bacteria like E. coli and S. aureus. For specific applications, more detailed studies are needed using specific strains. To develop coatings for medical implants, it is necessary to conduct more detailed studies in vivo.

Finally, it is necessary to briefly discuss the prospects for practical application of the ALD technologies and approaches described in this review. Most of the work presented requires additional reproducibility testing as well as additional in vitro and in vivo studies, some of the work requires investigation of a number of other functional properties. Nevertheless, many of the coatings obtained can be used in practice and industry after careful evaluation of their cost effectiveness. Although ALD has already become an industrial method in microelectronics and is even successfully used in the medical industry for surface modification of titanium implants [140,141], ALD is still a very expensive and complex technology with a very low coating growth rate. However, technical advances in recent years suggest that the impact of these negative factors can be reduced. In particular, we should note the development of technically simpler technologies such as atmospheric pressure ALD [142,143] and faster roll-to-roll ALD, spatial ALD [19], as well as the emergence of techniques that avoid the use of plasma to produce silver [123,144]. We believe that the advancement of these technologies could greatly augment the likelihood of extensive application of ALD in the manufacturing of antibacterial-coated materials within the industry.

Reviewer 3 Report

Comments and Suggestions for Authors

Title:

  Atomic layer deposition of antibacterial nanocoatings: a review  

Authors:

  Denis Nazarov, Lada Kozlova, Elizaveta Rogacheva, Ludmila Kraeva, and Maxim Maximov

This review details the results of using ALD for the synthesis of antibacterial coatings based on metals and oxides. The results of the analysis showed that the ALD method can be successfully applied to obtain a variety of different antibacterial nanocoatings on various inorganic and polymeric substrates. The influence of coating preparation conditions and their properties such as thickness, crystal structure, morphology, surface charge, wettability, etc. on the antibacterial properties was discussed in detail.

But atomic layer deposition of antibacterial nanocoatings as the significant approach against bacterial pathogens has only focused on S. aureus and E. coli. There are few other bacterial pathogens.

Would you please adding more references to state their antibacterial activity against other bacterial pathogens?

In addition, there are some grammar and writing errors.

Comments on the Quality of English Language

Minor editing of English language required

Author Response

This review details the results of using ALD for the synthesis of antibacterial coatings based on metals and oxides. The results of the analysis showed that the ALD method can be successfully applied to obtain a variety of different antibacterial nanocoatings on various inorganic and polymeric substrates. The influence of coating preparation conditions and their properties such as thickness, crystal structure, morphology, surface charge, wettability, etc. on the antibacterial properties was discussed in detail.

But atomic layer deposition of antibacterial nanocoatings as the significant approach against bacterial pathogens has only focused on S. aureus and E. coli. There are few other bacterial pathogens.

Would you please adding more references to state their antibacterial activity against other bacterial pathogens?

Thank you for your question. In most cases, researchers use S. aureus and E. coli as they are the most available, common and standardised strains of Gram-positive and Gram-negative bacteria. In addition, E. coli and S. aureus are the predominant causative agents of the majority of healthcare-associated and nosocomial infections. In some cases, a more specific C. albicans pathogen is used if the authors plan to use the coatings in specific areas such as catheters or contact lenses. Oral bacteria such as S. sanguinis, Bifidobacterium and Porphyromonas gingivalis are often used for antibacterial dental coatings. We have discussed these cases in section "4. Applications for ALD antibacterial coatings". Indeed, atomic layer deposition allows the synthesis of antibacterial coatings that are effective against various pathogens, as shown in our review. In addition, we discuss the effect of coatings on different strains and viruses in the sections on TiO2 and ZnO: "3.1.6. Effect of coatings on different strains" and "3.3.6. Effects on different types of bacteria and viruses". Unfortunately, such a discussion is not possible for ALD silver as there are very few studies and only S. aureus has been used.

We have presented all available information on ALD coatings and strains and it is not possible to add anything more. We have included a discussion on the limitations of the antibacterial study, which only utilized E. coli and S. aureus, in section 5.2 (the penultimate paragraph).

In addition, there are some grammar and writing errors.

The text of the manuscript was double-checked, grammatical and writing errors were corrected.

Round 2

Reviewer 1 Report

Comments and Suggestions for Authors

Publish as is.

Author Response

Thank you for your comments and evaluation of our manuscript!

Reviewer 2 Report

Comments and Suggestions for Authors

Authors have responded well to my previous questions. Only one more point should be cleared further, i.e. the category number in SI should start from "1", not directly copy from the previous manuscript and paste into the current SI. 

Author Response

Thank you for your comment! We have changed the numbering from SI.

Reviewer 3 Report

Comments and Suggestions for Authors

Title:

Atomic layer deposition of antibacterial nanocoatings: a review 

Authors:

Denis Nazarov, Lada Kozlova , Elizaveta Rogacheva, Ludmila Kraeva, and Maxim Maximov 

The manuscript can be accept in the present form.

Comments on the Quality of English Language

Moderate editing of English language required.

Author Response

Thank you for your comments and evaluation of our manuscript.